# A Y-linked duplication of anti-Mullerian hormone is the sex determination gene in threespine stickleback

Matthew J. Treaster⬤, Jenny McCann⬤, Kyra S. Solovei, Ryan J. Palmieri, Michael A. White⬤*

Department of Genetics, University of Georgia, Athens, Georgia, United States of America

* whitem@uga.edu

## Abstract

Many taxa have independently evolved genetic sex determination where a single gene located on a sex chromosome controls gonadal differentiation. The gene anti-Mullerian hormone (*amh*) has convergently evolved as a sex determination gene in numerous vertebrate species, but how this gene has repeatedly evolved this novel function is not well understood. In the threespine stickleback (*Gasterosteus aculeatus*), *amh* was duplicated onto the Y chromosome (*amhy*) ~22 million years ago. To determine whether *amhy* is the primary sex determination gene, we used CRISPR/Cas9 and transgenesis to show that *amhy* is necessary and sufficient for male sex determination, consistent with the function of a primary sex determination gene. We find that *amhy* contributes to a higher total dosage of *amh* early in development and likely contributes to differential germ cell proliferation key to sex determination. The creation of sex-reversed lines also allowed us to investigate the genetic basis of secondary sex characteristics. Threespine stickleback have striking differences in behavior and morphology between sexes. Here we show one of the classic traits important for reproductive success, blue male nuptial coloration, is controlled by both sex-linked genetic factors as well as hormonal factors independent of sex chromosome genotype. This research establishes stickleback as a model to investigate how *amh* regulates gonadal development and how this gene repeatedly evolves novel function in sex determination. Analogous to the "Four Core Genotypes" model in house mice, sex-reversed threespine stickleback offer a new vertebrate model for investigating the separate contributions of gonadal sex and sex chromosomes to sexual dimorphism.

### Author summary

Many species have evolved sex chromosomes like the XY system found in humans and other mammals. While sex chromosomes can contain hundreds of genes, a single sex determination gene found on the sex chromosomes controls

**Data availability statement:** RNAseq data is available in the NCBI SRA under BioProject PRJNA1248589. Raw and normalized read counts are available in the NCBI GEO under accession GSE296766. All scripts are available at https://github.com/MTreasterUGA/Amhy-sex-determination.

**Funding:** This study was supported by the National Science Foundation, MCB 1943283 to MAW, the National Institute of General Medical Sciences of the National Institutes of Health, R01GM147312 to MAW, the National Institute of General Medical Sciences of the National Institutes of Health, 1T32GM142623 to MJT, The University of Georgia Research Foundation to MJT, The Society for the Study of Evolution, R.C. Lewontin Early Award to MJT, and the ARCS Foundation, ARCS award to MJT. The content is solely the responsibility of the authors and does not necessarily represent the official views of the National Institutes of Health. The funders had no role in study design, data collection and analysis, decision to publish, or preparation of the manuscript.

**Competing interests:** The authors have declared that no competing interests exist.

whether the gonad develops into an ovary or testis. Many different sex determination genes have been identified across species, but we still have much to learn about how different sex determination genes control the same process of sex determination. Here, we show that an extra copy of the gene anti-Mullerian hormone on the Y chromosome of threespine stickleback fish is the sex determination gene responsible for initiating testis development. By manipulating this sex determination gene, we are now able to generate male and female stickleback with either XX or XY sex chromosome genotypes. This allows us to investigate the role of sex chromosomes in fitness and development outside of sex determination which is not possible in many models. Surprisingly, we find that the Y chromosome is not necessary for male fertility in stickleback. We also show that a key secondary sex characteristic, male mating color, is controlled both by hormones produced by the gonad as well as independent genetic factors on the sex chromosomes.

## Introduction

For species that have two discrete sexes, sex determination is the process that designates whether an individual will become male or female [1,2]. While this process is essential to the fitness and survival of a species, it is regulated by a surprising variety of initial cues and can incorporate a range of extrinsic signals as well as intrinsic genetic factors often located on sex chromosomes [1–3]. Numerous eukaryotic taxa have convergently evolved sex chromosomes, such as the XX-XY system found in mammals, to regulate this critical developmental process [1,2]. Sex chromosomes are derived from a pair of autosomes after one of these autosomes acquires a sex determining (SD) gene [4]. Suppression of recombination around this gene leads to sequence divergence of the two autosomes into a pair of divergent sex chromosomes [4]. Why this recombination suppression occurs is not clear; however, the prevailing model is that suppressed recombination will be selected for to link an SD gene with nearby sexually antagonistic mutations that are beneficial to one sex but detrimental to the other [4–6]. Empirically testing which genes on sex chromosomes are sexually antagonistic remains challenging, as these genes are perfectly linked to the sex determination gene. Thus, evaluating the fitness effects of sex-limited loci in the opposite sex (i.e., Y-linked loci in females) is not possible in most systems. Novel systems where sex can be isolated from other loci on the sex chromosome would be helpful for testing these theoretical models of sex chromosome evolution.

A wide variety of genes have evolved novel function as SD genes, but some genes appear predisposed to acquiring this role and have convergently evolved as SD genes in distantly related taxa [7]. *Sox3*, the ancestor of the mammalian SD gene, *Sry,* also acts as the SD gene in the Indian rice fish (*Oryzias dancena*) [8]. SD genes derived from *dmrt1* are found in birds [9], the African clawed frog (*Xenopus laevis*) [10], and medaka (*Oryzias latipes*) [11,12]. The most prevalent SD gene among vertebrates by far is *anti-Mullerian hormone* (*amh*) [7]. Independently acquired

Y-linked copies of *amh* or its dedicated receptor, *amhrII*, have been identified in at least twelve different clades of teleost fish [13–24] as well as in monotremes [25]. These Y-linked copies of *amh* and *amhrII* are presumed to be SD genes in all of these taxa. Although *amh* accounts for over one quarter of identified SD genes in vertebrates, how *amh* controls this critical developmental process and evolves novel function in sex determination is not clear [7].

*Amh* is a member of the transforming growth factor beta (TGF-β) family of proteins that are shared across vertebrates (Reviewed in [26,27]). *Amh* is unique among TGF-β hormones as it has only one identified type II receptor, *amhrII*, of which *amh* is the only known ligand [28–30]. *Amh* and its receptor were first identified for their role in reproductive tract development in mammals, where secretion of Amh by the developing testes causes regression of the primordial female reproductive tract, the Müllerian ducts [31]. Disruption of *amh* signaling in teleosts, which lack Müllerian ducts, causes overproliferation of germ cells in the gonad and can interfere with sex determination, even in species were *amh* is not the SD gene [32–34]. This suggests the ancestral function of *amh* is likely in germ cell proliferation and maintenance which, if altered, can disrupt sex determination, but how this disruption occurs is not clear. It also is not known whether Y-linked copies of *amh* that have convergently evolved function in sex determination control the downstream differentiation network in the same way or if *amh* can regulate sex differentiation through a variety of distinct mechanisms.

Species of stickleback fish (family Gasterosteidae) have multiple, independently evolved sex chromosome systems, making them an ideal model for investigating the origins of sex chromosomes and SD genes [20,35–38]. Two separate duplications of *amh* that are associated with male sex have been identified within stickleback fish. In the genus *Gasterosteus*, *amh* was duplicated from autosome 8 onto autosome 19 approximately 22 million years ago (mya), leading to the evolution of a Y chromosome [20,39]. An independent duplication of *amh* onto chromosome 20 was identified in the brook stickleback, *Culea inconstans*. This duplication differs from the ancestral copy on autosome 8 by only a handful of single nucleotide polymorphisms (SNPs), suggesting this duplication occurred much more recently than in the genus *Gasterosteus* [22,40]. Consistent with its recent acquisition, the duplicate copy of *amh* is not a fully penetrant SD gene, and its association with male sex varies across populations [40]. This provides a unique opportunity to examine the convergence evolution of *amh* as an SD gene at two dramatically different stages of sex chromosome evolution.

The threespine stickleback (*Gasterosteus aculeatus*) is a powerful emerging model for evolutionary and developmental biology with a robust set of genomic and functional genetic tools [41,42]. Their small size, large clutch sizes, and relatively short generation time make them a highly tractable model for studying the developmental mechanisms of *amh* as an SD gene. Threespine stickleback also have many well-studied sexually dimorphic traits and mating behaviors that have provided substantial insight into the evolution of sexual dimorphism and its effect on fitness [43–64]. Like other members of its genus, threespine stickleback have retained a functional copy of autosomal *amh* on chromosome 8 as well as the Y-linked *amhy*. The AMH prodomain and TGF-β signaling domain within *amhy* are conserved (S1 Fig), indicating the Y-linked duplicate is likely functional [20]. In addition, *amhy* is expressed in larvae around the time of sex determination, consistent with a role in initiating male development [20]. Beyond this, it is unknown whether *amhy* has a functional role in male sex determination. In this study, we show that *amhy* is both necessary and sufficient for male sex determination in the threespine stickleback, demonstrating that it is the SD gene in this species. We also perform RNA-seq of stickleback embryos and larvae to characterize expression of *amhy* and early transcriptional differentiation of males and females. Our findings establish stickleback fish as a premier model to study both the molecular mechanisms of *amh* based sex determination as well as the convergent evolution of *amh* as a SD gene in vertebrates.

## Results

### CRISPR/Cas9 knockout of *amhy* caused male-to-female sex reversal in XY fish

If *amhy* is the SD gene in threespine stickleback, it should be both necessary and sufficient for male sex determination. To test this, we first knocked out function of *amhy* in XY fish using CRISPR/Cas9. We designed two sgRNAs targeting exon 1 and exon 3 of *amhy* (Fig 1). We injected stickleback embryos from lab-reared fish derived from two independent marine

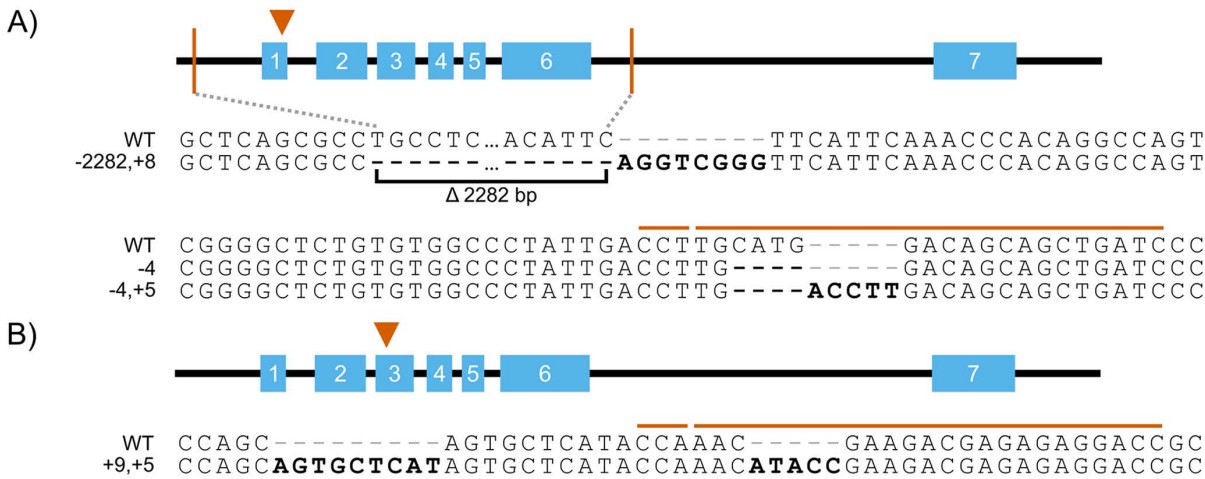

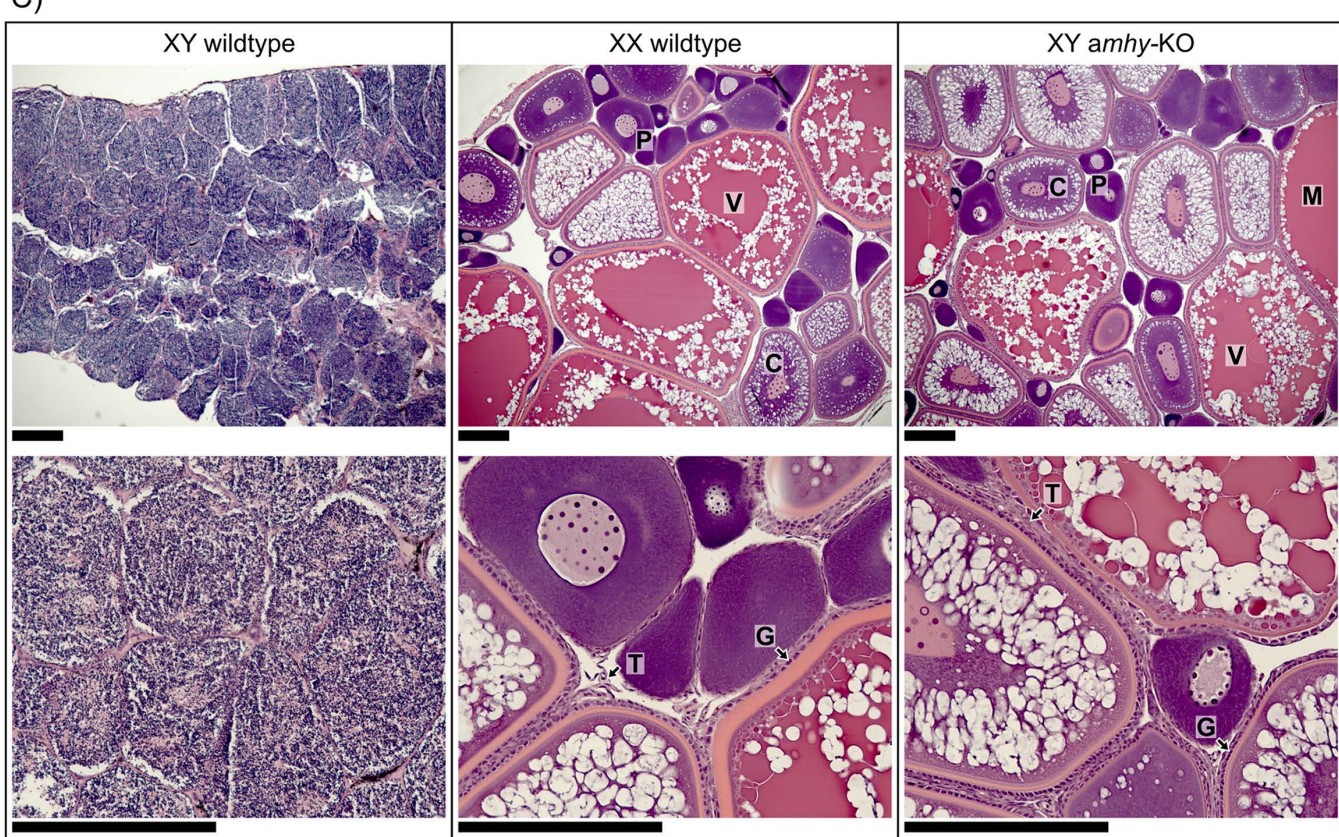

**Fig 1. CRISPR/Cas9 knockout of *amhy* caused male-to-female sex reversal in F1 XY fish.** We used CRISPR/Cas9 to target exon 1 (A) or exon 3 (B) of *amhy*. CRISPR target site and PAM sequence are indicated by orange lines above wildtype reference sequence. Mutations relative to wildtype are bolded. We identified four loss of function *amhy* alleles in F1 progeny. One allele (-2282,+8) deleted the first six exons of *amhy*. Other alleles featured small frameshift indels. C) We identified male-to-female sex reversal using gonad morphology and histology. Ovaries in XX wildtype and XY *amhy*-KO fish feature oocytes throughout multiple stages of development. P – primary oocyte, C – cortical alveolus stage, V – vitellogenic stage, M – Mature oocyte, G – Granulosa cells, T – Theca cells. Scale bars = 200 μm.

populations (Port Gardner Bay and Japan Pacific Ocean) with CRISPR ribonucleoprotein (RNP) made with one of the two sgRNAs. 8/17 (47%) of F0 XY fish injected with RNP exhibited male-to-female sex reversal. For all eight XY sex-reversed fish, we observed ovaries with all cell types seen in wildtype ovaries (S2A Fig).

F0 crispants are often mosaic for mutations induced by CRISPR/Cas9 gene editing (S2B Fig), which can cause variation in observed phenotypes. To generate non-mosaic F1 gene-edited fish, we used two F0 XY males that had heritable germline mutations in *amhy* but were not sex reversed, indicating they likely did not have *amhy* mutations in cell-lineages important for sex determination. Sperm from these fish were used to fertilize eggs from wildtype XX females. F1 offspring from these crosses inherited frameshift mutations in *amhy*, identified by Sanger sequencing (S3 Fig). We analyzed five F1 males from one F0 crispant with mutations in exon 1 and 21 F1 males from one F0 crispant with mutations in exon 3. All 26 F1 fish exhibited complete male-to-female sex reversal and developed ovaries morphologically and histologically indistinguishable from wildtype XX females (Fig 1C). No F1 XY fish showed development of testis or ambiguous gonad tissue with any visible spermatocyte production. We identified three loss of function alleles in exon 1 of *amhy* (Fig 1A) and one in exon 3 (Fig 1B). In ten F1 fish, the region of *amhy* in exon 3 targeted by CRISPR/Cas9 could not be amplified. PCR of exons 1 and 7 showed the presence of these distal portions of *amhy*, but PCR primers spanning a large putative deletion all failed to amplify, suggesting a chromosomal rearrangement disrupted *amhy* in these fish. As a specific mutation could not be positively identified, these samples were excluded from further analysis and final counts.

We analyzed the expression of canonical male differentiation genes *amh* and *dmrt1* and female differentiation genes *foxl2* and *cyp19a1a* using *in situ* hybridization on wildtype and F1 XY *amhy*-KO ovaries (Figs 2 and S4). We also

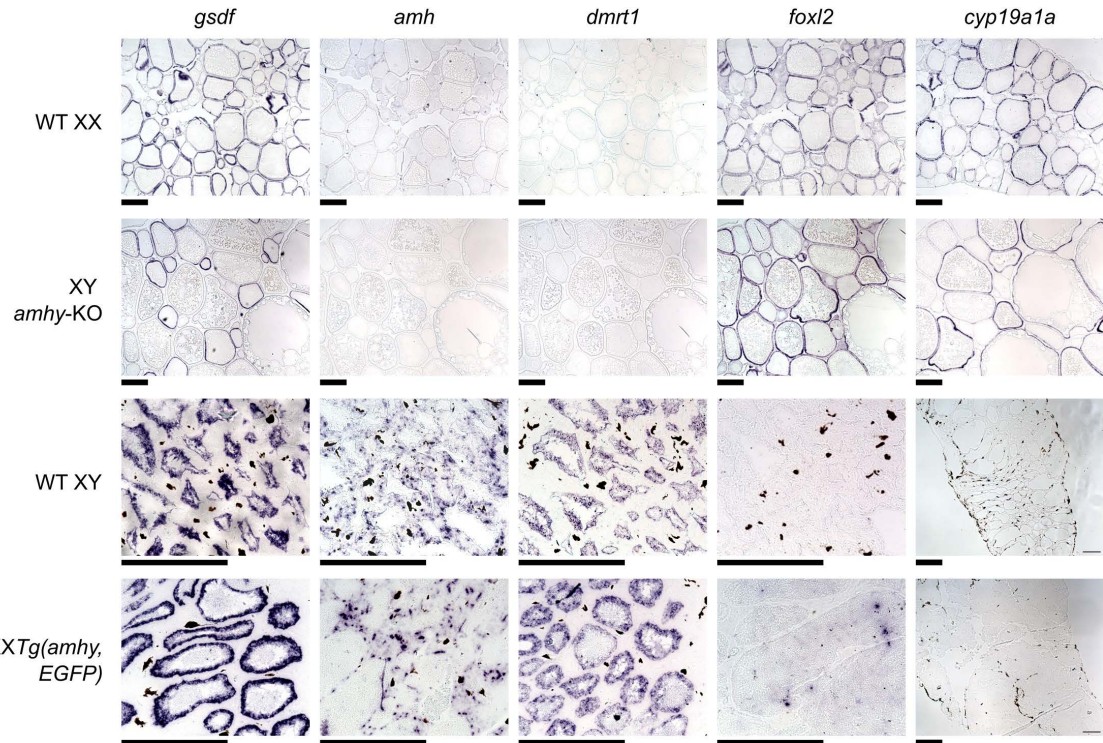

**Fig 2. Gene expression in sex-reversed gonads resembles expression in wild type gonads.** *In situ* hybridization of gonadal genes in wildtype and sex-reversed stickleback gonads. A positive control gene, *gsdf,* was expressed in somatic cells of both testes and ovaries. Wildtype XY and sex-reversed XX testes show strong expression of the male differentiation genes *amh* and *dmrt1* but not the female differentiation genes *foxl2* and *cyp19a1a*. Wildtype XX and sex-reversed XY ovaries strongly express female differentiation genes but not male differentiation genes. Scale bars=200 µm.

analyzed *gsdf* as a positive control, as this gene is expected to be expressed in somatic gonadal cells of both sexes [65]. Both wildtype and sex-reversed ovaries showed expression of *foxl2* and *cyp19a1a* in somatic cells. Expression of *amh* and *dmrt1* was minimal and restricted to oocytes when present as has been observed in some species of fish [21,66–69].

Knockout of *amh* or *amhrII* in teleost species without amh-based sex determination has caused male-to-female sex reversal, including in lab strains of zebrafish which lack sex chromosomes (reviewed in [70]) and medaka which have an XX-XY system with a *dmrt1* derived SD gene [32]. Thus, it is possible that off-target mutations of the paralogous *amh* could contribute to the sex reversal observed in F1 *amhy* mutants. Both sgRNAs have multiple mismatches to their homologous region in *amh*, and we verified that all F1 XY *amhy*-KO females did not have mutations within *amh* by Sanger sequencing (S5 Fig). This demonstrates that the sex reversal phenotype was not caused by mutations of *amh*. We also investigated whether the sex reversal phenotype was caused by a mutation at an unknown autosomal locus. Such an autosomal mutation must be dominant since we observed male-to-female sex reversal in heterozygous XY F1 fish. F1 XX siblings of sex-reversed XY fish would also be heterozygous for this mutation, and half of their F2 XY progeny would be sex reversed. We crossed two F1 XX females with wildtype XY males and phenotyped F2 offspring for sex reversal. All 16 F2 XY fish were phenotypically male, indicating that sex reversal in F1 XY *amhy*-KO fish was not caused by an off-target autosomal mutation.

## Introduction of an *amhy* transgene caused female-to-male sex reversal in XX fish

To determine if *amhy* is sufficient for male sex determination, we created a construct (pT2-5amhy2.5,Xla.Ef1a:EGFP) containing an 11 kb region of the Y chromosome which includes *amhy*, 5 kb of upstream sequence, and 2.5 kb of downstream sequence (S6 Fig). By including the surrounding non-coding sequences, we aimed to recapitulate the endogenous expression patterns of *amhy.* An Xla.Ef1a:EGFP cassette downstream of *amhy* was included to facilitate the screening of injected embryos for transgene integration (Fig 3C). These components were flanked by *Tol2* transposon sequences to facilitate random integration into the genome by the Tol2 transposase [71,72]. 16 F0 XX fish showed positive GFP expression and successful transgene integration, detected through PCR genotyping (Fig 3D). At reproductive age, all 16 F0 XX *Tg(amhy,EGFP)* were female-to-male sex reversed, developing testes rather than ovaries (Fig 3A). All 10 control F0 XX fish injected with the control Xla.ef1a:EGFP transgene showed positive presence of the transgene but did not show any signs of sex reversal and developed cytologically normal ovaries (Fig 3B). We also analyzed the expression of *amh, dmrt1, foxl2, cyp19a1a,* and *gsdf* using *in situ* hybridization on wildtype and F0 XX *Tg(amhy,EGFP)* testes (Figs 2 and S7). Both wildtype and sex-reversed testes showed strong expression of *amh* and *dmrt1* in somatic cells and no discernible expression of *foxl2* or *cyp19a1a*.

We attempted to constitutively express *amhy* using a synthetic exon-only coding sequence with an N-terminus FLAG tag joined to a 3' *EGFP* by a P2A self-cleaving peptide [73,74]. Plasmids were made using the *D. rerio* ubiquitin promoter [75] or the *X. laevis* ef1a promoter [76] to drive expression of the *amhy,EGFP* construct (S6 Fig). Transgenes with the respective promoter driving expression of only *EGFP* were used as controls. Somatic EGFP expression was observed with both control transgenes indicating these commonly used vertebrate constitutive promoters can drive expression in stickleback. Neither the Dre.Ubi:*amhy,EGFP* nor the Xla.Ef1a:*amhy,EGFP* transgene produced any larvae with visible EGFP expression, indicating that the *amhy* transgenes were failing to integrate or be actively expressed in any embryos. *Amh* has been overexpressed successfully in mouse [77], chicken [78], and Nile tilapia [16], and the P2A peptide has been successfully used in stickleback overexpression constructs [79], indicating that overexpression alone or the presence of a P2A peptide was likely not responsible for the failure of this construct. Introns of *amhy* may contain essential functional elements needed for gene expression of *amhy* in threespine stickleback. The presence of introns has been shown to impact gene expression, even under the regulation of non-native constitutive promoters [80,81].

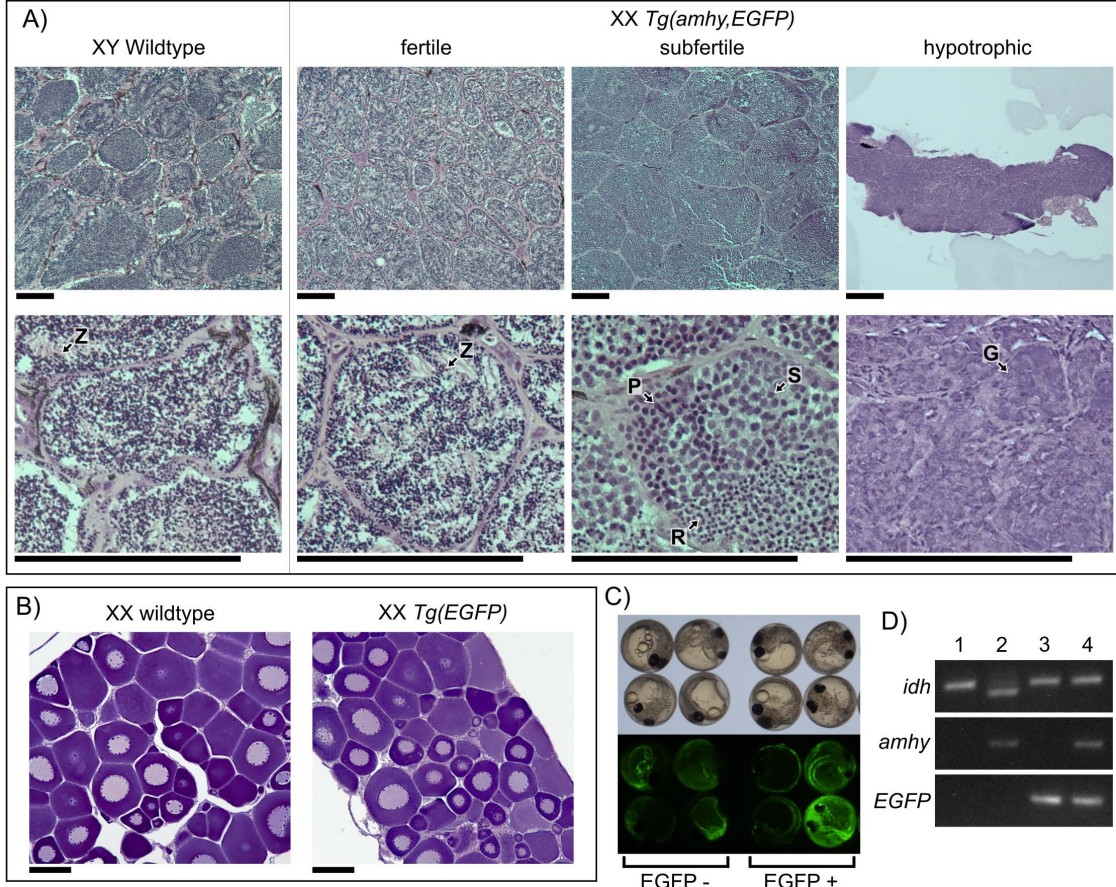

**Fig 3. XX fish with an *amhy* transgene exhibit female-to-male sex reversal.** A) XX *Tg(-5amhy3.5,Xla.Ef1a:EGFP)* fish developed testes with varying cell composition. Fertile fish had testes with clearly defined cysts of mature sperm as is found in wildtype fish. The subfertile testis had meiotic spermatocytes and immature round spermatids while hypotrophic testes had only undifferentiated spermatogonia. G – spermatogonia, P – primary spermatocyte, S – secondary spermatocyte, R – round spermatid, Z – Spermatozoa Scale bars = 100 μm. B) Control XX *Tg(Xla.Ef1a:EGFP)* fish developed ovaries identical to wildtype XX counterparts. Scale bars = 100 μm. C) Embryos injected with pT2-5amhy3.5,Xla.Ef1a:EGFP seven days after fertilization under bright field (top) and EGFP excitation (bottom). Embryos positive for transgene integration show somatic EGFP expression. We removed embryos with no EGFP expression or EGFP expression only in the yolk, which indicates the yolk was injected rather than the blastomere [41]. D) We used PCR of the sex-linked marker *idh* to genotype XX and XY fish. PCR of EGFP and *amhy* show the presence of the associated transgene. 1 – wildtype XX, 2 – wildtype XY, 3 – XX *Tg(EGFP)*, 4 – XX *Tg(amhy,EGFP)*.

## Sex-reversed *amhy* mutants produce viable gametes but are subfertile

Female-to-male sex-reversed F0 XX fish had varied fertility. Four of the 16 XX *Tg(amhy,EGFP)* males had testes morphologically resembling those of a fertile, wildtype male. 12 XX males had abnormally small and underdeveloped testes. We performed crosses with all four males with mature testes with wildtype XX females. We identified viable, successfully fertilized embryos by the presence of a gastrula and active epiboly 12–24 hours post fertilization (hpf). One male was extremely subfertile and only 3% of embryos were viable (n = 911). The remaining three males showed normal viability rates (80% n = 97; 94% n = 89, 98% n = 235), similar to viability rates of three wildtype males (73% n = 314, 87% n = 252, 97% n = 291) (S1 Table). One fertile male also produced offspring with somatic EGFP expression at a very low frequency (7%, n = 231), indicating a heritable germline integration of the transgene containing the constitutive Ef1a:EGFP cassette had occurred. Histology of F0 XX *Tg(amhy,EGFP)* males showed key differences between fertile, subfertile, and

hypotrophic testes (Fig 3A). Fertile males had normal testis histology with visible mature spermatozoa. The subfertile male testis lacked spermatozoa, instead containing meiotic spermatocytes and immature round spermatids concordant with its reduced fertility. Histology of the hypotrophic testes showed the germ cells were still early undifferentiated spermatogonia. This phenotypic variation may be a result of variable *amhy* expression influenced by transgene copy number and insertion location.

Sex-reversed XY *amhy*-KO females produced mature oocytes and became gravid, but they were subfertile compared to wildtype females. To assess fertility, we crossed three F1 XY *amhy*-KO females to wildtype XY males. These clutches all showed extremely low viability rates (2% n = 43, 4% n = 51, and 6% n = 63) (S2 Table). One of the three F1 fish was crossed four additional times and showed improvement in viability with the subsequent crosses. The viability rates of the fourth and fifth cross were 78% (n = 100) and 79% (n = 89) respectively, though this is still lower than the average viability of eggs from nine wildtype females (85% n = 857) (S1 and S2 Tables). Three days after fertilization, 50 of the 148 viable F2 eggs had arrested development. We were able to genotype 48 of the failed embryos. 36 of the 48 arrested embryos were YY and carried both the paternally transmitted Y chromosome (Y$^+$) and the maternally transmitted *amhy*-KO Y chromosome (Y$^{KO}$) (Fig 4D). The remaining embryos were reared to adulthood and 92 were genotyped for their sex chromosome complement. No adult Y$^+$Y$^{KO}$ fish were observed, indicating that the YY genotype is embryonic lethal due to the missing X chromosome (Fig 4A and 4B). XY$^{KO}$ females were found to transmit the X and Y$^{KO}$ chromosome with equal frequency, indicating there was not any biased transmission of the X or Y chromosome during female meiosis (36 Y$^+$Y$^{KO}$ embryos out of 148 total does not deviate from a 1:3 Mendelian ratio; chi-squared = 0.0360; p = 0.8494; of the 92 adult F2s genotyped 36 XX, 27 XY$^+$, and 29 XY$^{KO}$ does not deviate from the expected 1:1:1 ratio; chi-squared = 1.457; p = 0.4827). We also measured fertility in an F2 cross which contained sex-reversed XY$^{KO}$ females and wildtype XX female siblings. We crossed 11

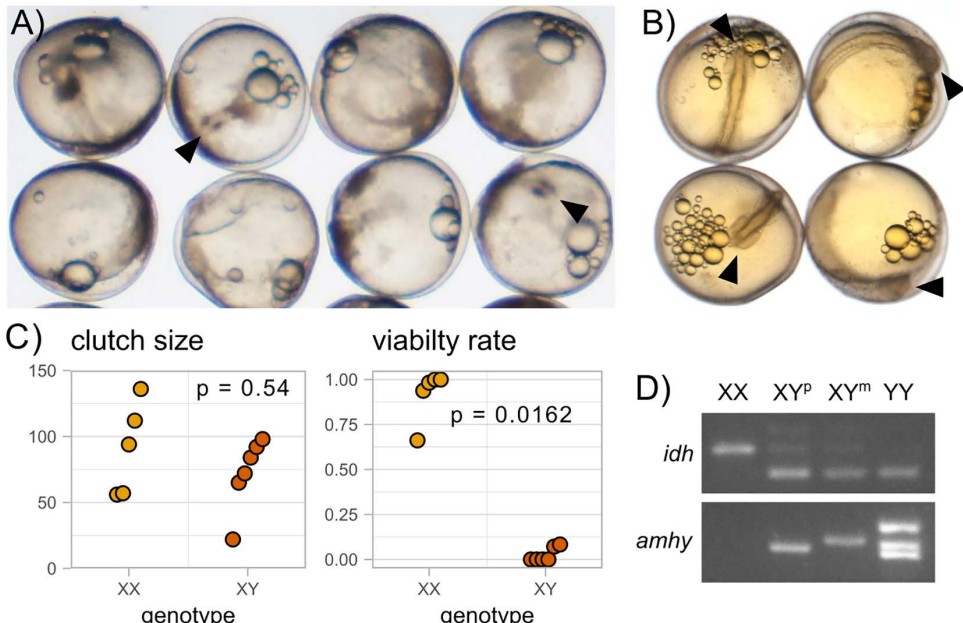

**Fig 4. XY *amhy*-KO females can reproduce and transmit an X or Y chromosome.** YY embryos (A) at 48 hours post fertilization show arrested development compared to wildtype embryos (B). Arrowheads denote anterior end of body axis. C) F2 XX and XY *amhy*-KO female siblings do not show a significant difference in clutch size (Wilcoxon test, p = 0.54). Clutches from XY *amhy*-KO females have a significantly lower viability rate than those from XX female siblings (Wald z-test of beta regression coefficients, z = -2.405, p = 0.0162). D) PCR of the sex-linked marker IDH shows the absence of the X chromosome in YY embryos. Both the paternal (wildtype) and maternal (+9, +5) loss of function alleles can be detected in XY offspring. YY embryos show the presence of both amhy alleles as well as a third band, likely a heterodimer of the maternal and paternal amplicons.

F2 females (5 XX and 6 XY^KO) with wildtype XY males. The clutch size did not differ significantly between XY^KO and XX females (Fig 4C and S3 Table). However, the viability rate was substantially lower in the F2 XY^KO females, similar to the F1 generation (Fig 4C and S3 Table). Two F2 females were also eggbound and were unable to be stripped of eggs. Both eggbound females were XY^KO, and no eggbound XX females were observed at the time crosses were performed.

## Male nuptial coloration is present in both sex-reversed genotypes

Male nuptial coloration is one of the most prominent sexually dimorphic characteristics in threespine stickleback. During their breeding season, males from many stickleback populations develop red throats and dark blue eyes and bodies [82]. To investigate the separate contributions of gonadal and genotypic sex on this trait, we compared coloration in mature sex-reversed XX *Tg(amhy,EGFP)* males and XY *amhy*-KO females to wildtype XX and XY fish (Fig 5). Neither XX *Tg(amhy,EGFP)* males nor XY *amhy*-KO females appeared to fully recapitulate wildtype XY male coloration, but both were distinct from wildtype XX females (Fig 5A). XX males had blue coloration on their eyes and bodies, but none were as dark as completely colored XY males. XY females were varied in appearance, but some had discernibly blue eyes and darkened bodies not seen in XX females. We measured the proportion of body area that was blue or dark for each genotype (S8 Fig and S4 Table). As red throats are not prominent even in wildtype males under our laboratory conditions, we were

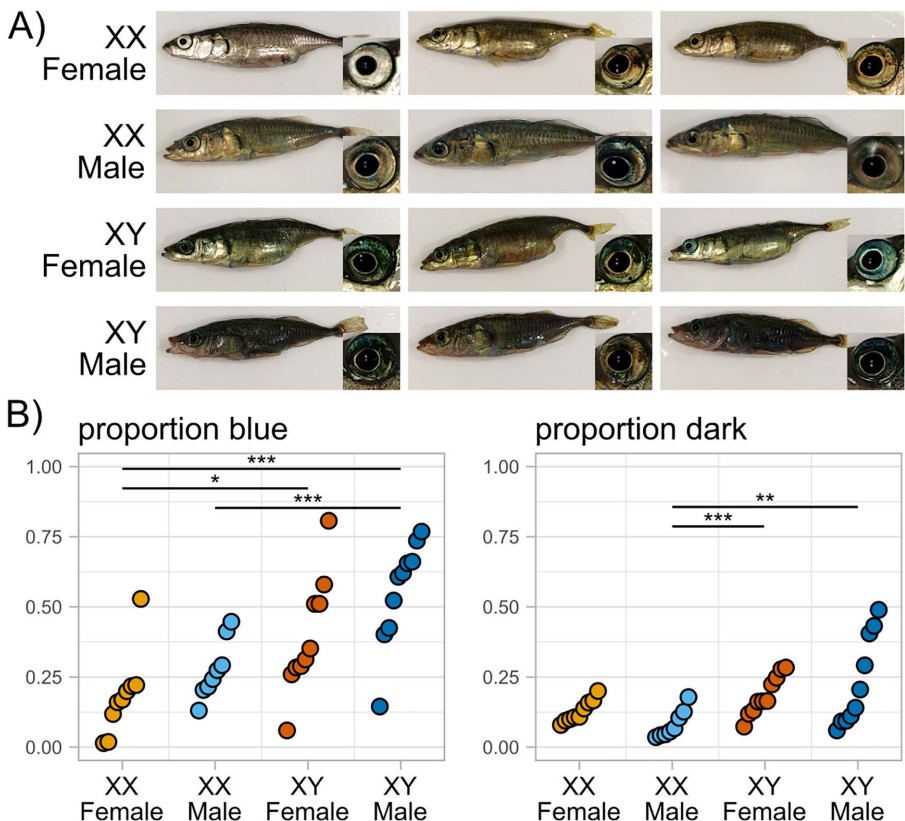

**Fig 5. Both male and female sex-reversed stickleback show partial male nuptial coloration.** A) Representative images of wildtype XX females and XY males and sex-reversed XX *Tg(amhy,EGFP)* males and XY *amhy*-KO females. XX females are silver and may darken slightly when gravid. XY males have dark blue eyes and bodies and red throats. Sex-reversed XX males and XY females show intermediate aspects of male nuptial coloration. B) Proportion of body area that is blue or dark for each genotype. Differences in coloration between genotypes were tested using estimated marginal means from a beta regression with Wald z-tests and Tukey adjustment for pairwise comparisons (* p<0.05, ** p<0.01, *** p<0.001) (S4 Table).

not able to quantify this trait. Both proportion of dark and blue body area showed wide variation within genotypes (Fig 5B). We saw significantly more blue (p < 0.0001) but not dark coloration (p = 0.1161) in XY males compared to XX females. XX males were significantly less blue (p = 0.0003) and dark (p = 0.0007) than XY males. XY females were significantly more blue than XX females (p = 0.0348), but there was no difference in darkness (p = 0.1074). Together, these findings show that male nuptial coloration is influenced both by hormones produced by the testis and by sex chromosome genotype, but both male gonad differentiation and an XY genotype is needed to present full male nuptial coloration.

### *amhy* contributes to higher total *amh* dosage early in development

Since Amh shows high specificity for its dedicated receptor, AmhrII, [28–30] we predicted that *amhy* would be functionally constrained regarding coding sequence evolution and would instead evolve novel function through regulatory changes. We investigated the degree of sequence conservation within putative regulatory regions (2 kb upstream and downstream of the coding sequence), of *amhy* relative to autosomal *amh*. We also compared sequence similarity between *amh* and the orthologous *amh* in the ninespine stickleback (*Pungitius pungitius*) which do not have an *amhy* (Fig 6 and S5 Table). The divergence time between threespine and ninespine stickleback is approximately 26 mya [83], similar to the 22 mya divergence time between the threespine *amh* and *amhy* [20]. There was less sequence identity between the coding sequences of threespine *amh* and *amhy* (0.780) than between threespine *amh* and ninespine *amh* (0.879). Lower sequence identity despite shorter divergence time is consistent with weaker purifying selection acting on the Y chromosome due to a lack of recombination [84]. This was also supported by a higher overall dN/dS value between threespine *amh* and *amhy* (0.780) than between threespine *amh* and ninespine *amh* (0.668), though this pattern did vary across exons (S5 Table). Alternatively, reduced sequence identity and elevated dN/dS could be due to positive selection acting on coding sequence changes to *amhy*. The sequence identity of upstream and downstream regions was much lower between threespine *amh* and *amhy* (0.365 and 0.376) than between threespine *amh* and ninespine *amh* (0.595 and 0.660), indicating *amhy* has undergone significant regulatory evolution since its duplication onto the Y chromosome, which may cause differences between *amh* and *amhy* expression.

To determine whether expression of these genes differ throughout development, we sequenced the total transcriptomes from prehatching embryonic stages 17 (early somitogenesis, ~43 hpf), 20 (onset of pigmentation, ~73 hpf), and 23 (completion of yolk sac circulation, ~122 hpf), stage 26 larvae (0 days post hatching (dph), around 200 hpf), as well as larvae 4 and 8 dph [85]. The earliest identified difference between male and female threespine stickleback is an increase in primordial germ cell count in females relative to males at 3–4 dph, followed by differences in germ cell morphology at 7 dph [86].

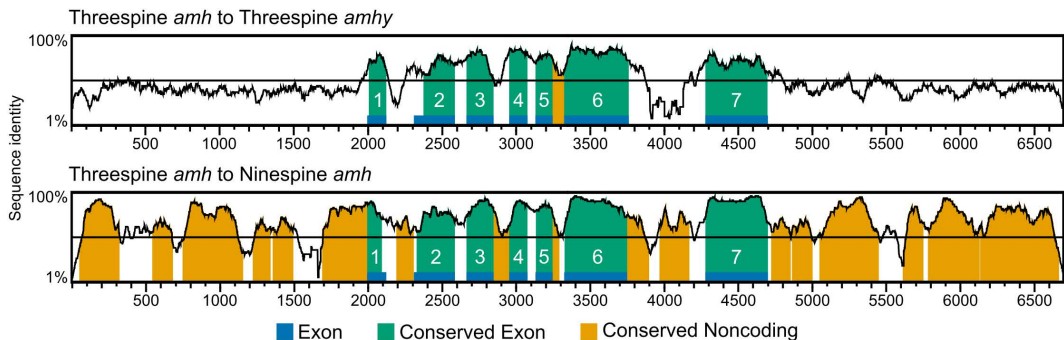

**Fig 6. *amh* and *amhy* show no sequence conservation outside of the coding sequence.** Vistaplots showing conserved sequence identity of threespine *amh* to threespine *amhy* (top) and to ninespine *amh* (bottom) with sequence identity on the y axis and nucleotide position of threespine *amh* on the x axis. Exon positions are indicated along the x axis in blue. Conserved exon and noncoding sequences share 70% sequence identity in a 100 bp window.

Sex determination should begin sometime prior to this stage. We predicted that the unique regulatory architecture of *amhy* resulted in earlier expression timing relative to *amh.* Thus, we predicted *amhy* would be expressed at higher levels relative to *amh* early in development in order to initiate sex determination. As the gonad differentiates, *amh* expression should increase relative to *amhy* due to the canonical role of *amh* in male differentiation. This pattern has been observed in Japanese flounder [15] and Patagonian pejerrey [13], which also have *amhy* SD genes. To compare *amh* and *amhy* expression, we first scaled *amhy* read counts by the difference in transcript length which is not accounted for by normalized read counts (S6 Table). We then tested whether *amhy* expression was different from one half of autosomal *amh*, as males have two copies of *amh* and one copy of *amhy*. Average expression of *amhy* was higher than one half *amh* at stage 23 and all timepoints post hatching, but this difference was only significant at stage 23 (p = 0.0454). We found *amhy* expression trended towards higher expression at later developmental stages, consistent with a role in sex determination. However, this difference was only significant between stage 20 and 4 dph (p = 0.0398, Tukey HSD = 0.0361). We then hypothesized that male sex determination could be initiated by a higher overall dose of *amhy* when expressed simultaneously with *amh.* We combined *amh* and transcript length adjusted *amhy* normalized read counts at each stage and found total amh dosage was higher in males at 0 dph (p = 0.0448), 4 dph (p = 0.0347), and 8 dph (p = 0.0297) (S6 Table), across the time where germ cell count diverges in males and females. Together, these results suggest that increased total dosage of *amh* provided by *amhy* is key to its role in male sex determination.

## Sex differentiation genes are not differentially expressed between males and females early in development

Following sex determination, we expected to see divergence in the expression of overall male and female transcriptomes. However, principal component analysis of autosomal gene expression indicated that male and female larvae had similar expression profiles. Autosomal genes were clustered by developmental time rather than separating by sex at any stage of development (Fig 7A). Only sex chromosomes showed clear separation by both sex and developmental time (S9 Fig), since the X chromosome is expressed two-fold higher in females, relative to males, and the Y chromosome is only expressed in males. The overall lack of sex-specific expression on the autosomes suggests that many aspects of sex differentiation occur well after the earliest noted phenotypic difference between males and females: the proliferation of germ cells in the primordial gonad at 3–4 dph [86].

We examined whether any developmental stages had genes with known functions in sex differentiation that were significantly differentially expressed between males and females. Many male and female differentiation genes are highly conserved across vertebrates [2] and should also be expressed during development of the threespine stickleback. For instance, we expected differential expression of key transcription factors canonically involved in differentiation, including the male biasing factors *dmrt1* and *sox9b* or the female biasing factors *foxl2a*, followed by divergence of downstream genes such as the aromatase enzyme *cyp19a1a* and the TGF-b hormones *amh* and *gsdf*. In total, we found 455 differentially expressed genes (DEGs) between males and females on the autosomes (S1 File). Key regulators of differentiation should be differentially expressed throughout development and into adulthood in order to actively maintain gonadal fate [2]. However, 430 of the DEGs were only differentially expressed in a single stage (S1 File), suggesting these genes do not play a substantial, ongoing role in sex differentiation. Additionally, while we expected to see an increase in the number of DEGs over time due to activation of male and female specific gene networks, the number of DEGs was highest at stage 17 (156 DEGs), decreasing in later stages until a small increase from 4 dph (41 DEGs) to 8 dph (65 DEGs) (S10 Fig). None of the canonical differentiation genes of interest were differentially expressed between males and females at any timepoint examined in this study (Fig 7 and Table 1). Male differentiation genes *dmrt1* and *amh* and female differentiation genes *foxl2* and *cyp19a1a* were differentially expressed in adult gonads of both wildtype and sex-reversed fish shown by *in situ* hybridization (Figs 2, S4, and S7), indicating their function in differentiation are conserved in stickleback. Of the identified DEGs, only a small handful have known associations with gonadal function, development, or differentiation. Two genes associated with germ cell specification and maintenance [87], *ddx4* and *piwil1*, showed female biased expression

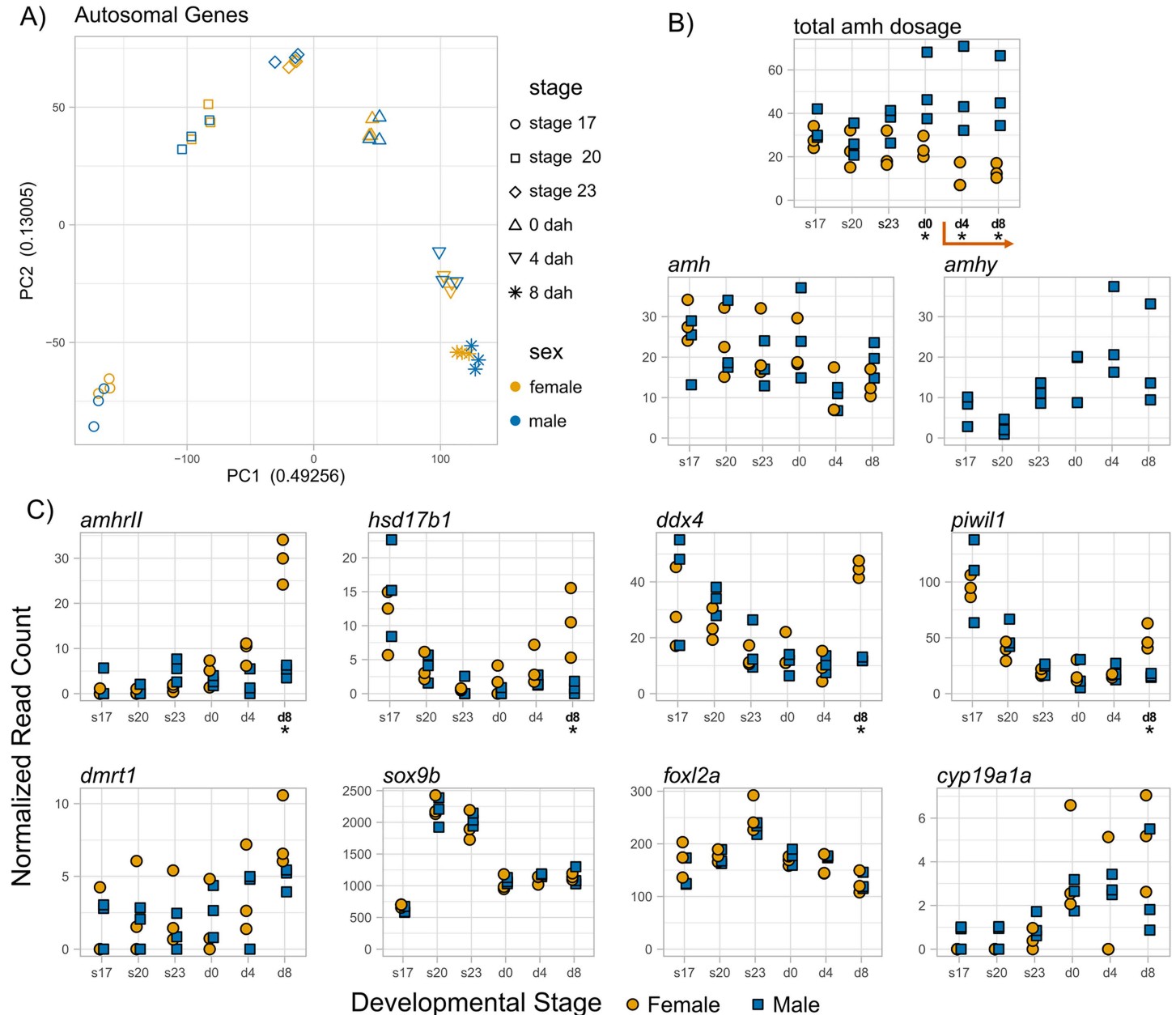

**Fig 7. Expression of gonadal growth and differentiation genes in embryonic and larval stickleback.** (A) PCA plots of normalized read counts for all genes cluster by stage rather than sex. (B) and (C) Normalized read counts of genes in embryonic stages 17, 20, 23, and larvae 0, 4, and 8 days after hatching. Stages with significant differences between males and females are marked with an asterisk and bolded (p<0.05; Tables 1 and S6). The orange arrow (B) indicates stages where differences in germ counts have been observed between males and females.

at 8 dph (Fig 7C). Interestingly, *amhrII,* which shows female biased expression in embryonic mouse gonads [88], was also expressed more highly in females at 8 dph. Although *cyp19a1a* was not differentially expressed, the steroidogenic enzyme *hsd17b1*, which plays a role in estrogen synthesis [89], showed female biased expression at 8 dph. This gene acts as the SD gene in *Seriola* fishes [90] and is upregulated in ovaries of Siberian sturgeon before the activation of *foxl2* or *cyp19a1a* [91]. A lack of differential expression in many of these key genes indicates that gonad differentiation occurs after 8 dph.

**Table 1.** Benjamini Hochberg adjusted P values for differential expression between males and females.

| Gene | Gene Id | Stage 17 | Stage 20 | Stage 23 | 0 dph | 4 dph | 8 dph |
|------|---------|----------|----------|----------|-------|-------|-------|
| *amh* | amh | 1 | 1 | 1 | 1 | 1 | 1 |
| *amhrII* | LOC120835457 | 1 | 1 | 0.341 | 1 | 0.609 | **0.0313** |
| *ddx4* | ddx4 | 1 | 1 | 1 | 1 | 1 | **0.00225** |
| *piwil1* | piwil1 | 1 | 1 | 1 | 1 | 1 | **0.00245** |
| *hsd17b1* | hsd17b1 | 1 | 1 | 1 | 0.873 | 1 | **0.0272** |
| *foxl2a* | foxl2a | 0.832 | 1 | 1 | 1 | 1 | 1 |
| *cyp19a1a* | LOC120811580 | 1 | 1 | 1 | 1 | 1 | 1 |
| *dmrt1* | dmrt1 | 1 | 1 | 1 | 1 | 1 | 1 |
| *sox9b* | LOC120827768 | 0.946 | 1 | 1 | 1 | 1 | 1 |
| *gsdf* | gsdf | 1 | 1 | 0.483 | 1 | 1 | 1 |
| *nr5a1* | nr5a1a | 1 | 1 | 1 | 1 | 1 | 0.983 |

We identified some other DEGs that have known reproductive functions in other vertebrates but do not have clear connections to sex differentiation and are only differentially expressed at a single timepoint. Gonadotropin-releasing hormone II receptor (LOC120811940) has known roles in the stimulating steroidogenesis in both testes and ovaries across mammals [92] and had male biased expression at stage 17 (logFC = 2.74, P = 1.6E-3). TGF-beta 3 hormone is expressed more highly in male mice during gonadogenesis [93], and a TGF-beta-3 proprotein-like gene (LOC120807995) was expressed slightly higher in females at stage 17 (logFC = -0.63, P = 0.042). *Piwil2*, another germ cell associated gene [87], was expressed higher in males at stage 20 (logFC = 1.71, P = 4.0E-3). *Cdc25b* is expressed at low levels in mouse ovaries and testes [94] and was male biased at stage 23 (logFC = 0.52, P = 0.012). Separin/separase (*espl1*) has known roles in mitotic and meiotic chromosome segregation [95] and was expressed at higher levels in 8 dph females (logFC = -0.56, P = 1.4E-4). Additional work will be necessary to understand what function, if any, these genes have in threespine stickleback gonad development.

## Discussion

### *amhy* is the sex determination gene in threespine stickleback fish

Our findings demonstrate that *amhy* is the SD gene in threespine stickleback, as it is both necessary and sufficient for sex determination (Table 2). Independently derived *amhy* genes have been functionally demonstrated to have a role in sex determination in three other clades of teleost; however, threespine stickleback offers a novel perspective as their *amhy* is more diverged, and genetic sex determination is more canalized in this species. Nile tilapia and Japanese flounder have only one and nine SNPs, respectively, that differentiate their X- and Y-linked copies of *amh* [15,16], and *amhy* in Patagonian pejerry shares 94% nucleotide identity with its autosomal paralog [13]. In all three of these species, genetic sex determination can be overridden to varying degrees by temperature [96–98]. In contrast, the threespine stickleback, *amh* and *amhy* only share 78% nucleotide identity (S5 Table). There is no evidence that temperature can disrupt sex

**Table 2.** Phenotypes of mutant stickleback from this study.

| Genotype | N | Male (%) | Female (%) |
|----------|---|----------|------------|
| F0 XY *amhy* crispants | 17 | 9 (53%) | 8 (47%) |
| F1 XY *amhy*-KO | 16 | 0 (0%) | 16 (100%) |
| F0 XX *Tg(-5amhy2.5,Xla.Ef1a:EGFP)* | 16 | 16 (100%) | 0 (0%) |
| F0 XX *Tg(Xla.Ef1a:EGFP)* | 10 | 0 (0%) | 10 (100%) |

determination in threespine stickleback, and occurrences of sex-reversed or intersex stickleback in wild populations is rare [99,100]. This indicates that genetic sex determination is highly canalized in threespine stickleback and less susceptible to environmental signals than many other teleost species, making stickleback a robust model to investigate how *amh* regulates gonadal development. Comparative studies across these taxa will be invaluable for understanding how *amh* repeatedly evolves this novel function, and the presence of an independently derived male associated *amh* duplication in the closely related brook stickleback [22,40] makes the stickleback family an especially powerful tool for studying the convergent evolution of *amh* as an SD gene.

## Potential mechanisms of sex determination in stickleback

Germ cell density has been shown to affect sex determination in several teleost species, with the male fate associated with lower germ cell count and the female fate with a higher count. Common lab zebrafish strains are undifferentiated gonochorists, where all individuals begin ovary development, but the gonad transitions to testis development in some individuals [101]. Complete ablation of germ cells causes zebrafish to undergo male development [102], and a threshold of germ cells at the time of gonadal differentiation is necessary to sustain ovary development [103]. *O. latipes*, like sticklebacks, are differentiated gonochorists where either testis or ovary development begins from a common undifferentiated primordium [104]. Ablation of germ cells in XX *O. latipes* causes female-to-male sex reversal [105], and overproliferation of germ cells in XY fish causes male-to-female sex reversal [106]. The association of germ cells with sex has also been observed outside of teleosts in the red eared slider turtle (*Trachemys scripta elegans*) which has temperature dependent sex determination. Embryos incubated at female promoting temperatures have higher germ cell counts than those incubated at male promoting temperatures, and depleting germ cells at intermediate temperatures biased the sex ratio towards males [107]. Even after primary sex determination, germ cell depletion in mice [77] and rats [108] leads to masculinization of the ovary, which produces Sertoli-like cells and seminiferous tubule-like structures before its eventual degradation. This suggests the presence of a widely conserved, ancestral mechanism in vertebrates by which germ cell density governs gonad identity. A new duplication of *amh* could increase overall *amh* dosage, reducing germ cell proliferation to co-opt this underlying mechanism and become a male determining gene.

Consistent with this mechanism, we found evidence that *amhy* contributes to higher *amh* dosage in post hatching stages where differences in germ cell proliferation between males and females are first observable in the developing gonad [86]. We also found increased expression of germ cell specific genes (e.g., *ddx4* and *piwil1*) in females at 8 dph, corroborating previous histological observations of increased germ cell proliferation in female stickleback gonads beginning at 3 dph [86]. Overexpression of *Amh* in mice [77,109] causes reduced germ cell count and gonad size in males and females, and loss of *Amh* alters primordial follicle recruitment in females [110]. Similar findings have also been observed in chicken [78,111]. Studies misexpressing *amh* in teleosts have primarily focused on sex determination; however, gonadal hypertrophy observed in *amh* knockouts in other teleosts [32,34] alongside gonadal hypotrophy with a lack of mature germ cells observed in some XX *Tg(amhy,EGFP)* males in this study demonstrate the conserved function of *amh* suppressing germ cell proliferation in stickleback and other teleosts.

Somatic cells may also be affected by *amhy* in other ways. In mice, low concentrations of Amh promote proliferation and growth factor production in somatic cells, but increased concentrations induce apoptosis [112]. Altering somatic cell proliferation could lead to differences in the levels of hormones and growth factors that they produce, shaping gonadal development. While Amh is not necessary for initial gonadal differentiation, it is important to maintaining Sertoli cell fate in fetal mice [113]. Estrogen production in somatic cells also plays a major role in sex differentiation and ovary development in teleosts (reviewed in [114,115]). In tilapia, Amh signaling directly inhibits *cyp19a1a* production via Smad transcription factors [116]. While we did not detect a difference in *cyp19a1a* expression between males and females in embryos or larvae, it is expressed more highly in adult ovaries than in testes, so *amhy* could play a reinforcing role later in development to suppress estrogen synthesis in males.

The evolutionary changes that have given rise to novel function of *amhy* in stickleback sex determination remain unclear. While we do see substantial sequence divergence in the putative regulatory regions of *amhy* relative to *amh*, we only detected a significant difference in expression at a single developmental stage. As we are unable to isolate embryonic gonadal tissue for expression analysis, our results in this study are limited to characterizing total expression in the whole embryo. This likely reduces our ability to detect differences in the expression of genes like *amh* and *amhy* which are restricted to a small population of cells. Alternative approaches, like single cell RNA-seq, spatial transcriptomics, or HCR *in situ* will be useful to survey gene expression in the early differentiating gonad. Still, we should not rule out the significance of coding sequence evolution for *amhy*. In Nile tilapia, *amhy* expression is lower than the X-linked *amh* throughout development, and misexpression of *amhy* but not *amh* induces male sex determination [16]. This implies that the single missense SNP between *amh* and *amhy* in tilapia is important to the function of *amhy* in sex determination. In Spotted knifejaw (*Oplegnathus punctatus*), a Y-linked copy of *amhrII* shows higher affinity for binding *amh* than its X-linked paralog [23]. In zebrafish which lack *amhrII*, *amh* may instead use the type II TGF-β receptor *bmpr2a* [117,118]. Mutations that increase the stability of Amhy, increase its affinity for the receptor, AmhrII, or promote interaction with a non-canonical receptor could also affect gonadal development. Manipulating *amh* and *amhrII* could provide further insight into the evolution of *amhy* and its mechanism of inducing male development. We would predict knockout of *amh* or *amhrII* to result in overproliferation of germ cells and some level of male-to-female sex reversal in XY fish as is seen in medaka *amhrII* mutants [32]. This would demonstrate that germ cell proliferation regulated by the canonical *amh* signaling pathway can influence sex in stickleback, making protein evolution or interaction with non-canonical receptors less parsimonious avenues by which *amhy* has evolved its novel function. Misexpression of *amh* and *amhy* using regulatory elements from the opposite gene (i.e., expressing the *amh* genic sequence with *amhy* regulatory elements) or non-native promoters could also elucidate whether coding sequence evolution is important to the function of *amhy*.

## Sex differentiation occurs later than anticipated

We observed no differential expression in canonical vertebrate sex differentiation genes, including *dmrt1, sox9b, foxl2a,* and *cyp19a1a.* While the order and timing of expression varies across taxa, these core genes are reliably differentially expressed between males and females before the onset of morphological gonadal differentiation [119–121]. These genes show expected dimorphic expression patterns in adult gonads, shown both by our *in situ* data as well as gonadal transcriptomic data [122]. This indicates that there is some delay between the initial establishment of germ cell differences between males and females and the later differentiation of the primordial gonad into either a testis or ovary. Expression analysis of whole embryos and larvae could obscure gonad specific changes in gene expression as previously discussed. However, we did identify differential expression of *ddx4* and *piwi1l* which are highly germ cell specific [87] as well as *hsd17b1* and *amhrII* which are not gonad specific [89,123] but show early female biased gonadal expression in other species [88,91]. This demonstrates some ability to detect sex differences in expression for significant genes. Female biased expression of *hsd17b1* precedes differential expression of these canonical differentiation genes in sturgeon, where *hsd17b1* is thought to increase estrogen, upregulating of *foxl2* and *cyp19a1a* in females [91]. In stickleback, female biased *hsd17b1* expression may be a key difference between males and females, initiating expression changes in canonical differentiation genes. Further analysis of gene expression at later timepoints will be necessary to determine if this is the case.

We propose that sex determination in threespine stickleback occurs in two phases. In the early critical period of sex determination, germ cell proliferation in the undifferentiated gonad diverges between males and females due to the presence of *amhy* in males. Then, during the sensitive period of gonadal differentiation, the difference in germ cell count between males and females dictates gonadal fate, marking the end of the sex determination and the beginning of differentiation. Unlike *Sry* in mammals which is expressed in a brief window and directly initiates gonadal differentiation [124], *amhy* in stickleback appears to be expressed throughout this early period of germ cell proliferation. These two phases

of germ cell proliferation followed by gonad differentiation are also present in other species with SD genes including Nile tilapia (reviewed in [125]), and medaka (reviewed in [104]) as well as species without SD genes such as red eared slider [107] where germ cell proliferation is affected by temperature. This suggests that this two-phase system of differential germ cell proliferation followed by gonadal differentiation is broadly conserved across vertebrates, but what combination of environmental and/or genetic factors influences germ cell proliferation has changed between taxa. The ability of *amh* to immediately integrate with this preexisting sex determination network through its existing function in regulating germ cell proliferation along with possible reinforcing roles in regulating somatic cell identity and estrogen synthesis would explain the prevalence of *amh* as an SD gene.

**Fertility of sex-reversed threespine stickleback**

Both XY *amhy*-KO females and XX *Tg(amhy,EGFP)* male sex-reversed fish were fertile in our study. Fertility of sex-reversed mutants is not uncommon in other teleost species; however, it is often observed in species with younger or less diverged sex chromosomes. In these instances, the X and Y chromosomes contain a similar complement of genes. In medaka, the Y specific region is 280 kb and the only functional gene it contains is the sex determining *dmrt1by* [12]. In Nile Tilapia, the sex determining region on linkage group 23 is 1.5 Mb and contains 51 annotated genes, including the sex determining *amhy* [126]. In both species, sex-reversed fish are fertile and YY offspring are viable [11,127]. In contrast, the threespine stickleback Y chromosome has a ~15 Mb non-recombining region [20]. Within this region, the Y chromosome has lost half of the almost 1200 ancestral gametologs, causing inviability of YY embryos, and has accumulated many Y-specific genes not present on the X chromosome [20]. Sex chromosome-specific gene content evolution could lead to improper expression in the opposite sex, affecting gametogenesis. Given this divergence, the ability of both XY *amhy*-KO females and XX *Tg(amhy,EGFP)* males to reproduce is surprising. In mouse, reduced fertility of XY females is a result of meiotic errors between the X and Y chromosomes as well as the effect of aberrant Y-chromosome genes in the ooplasm on early development (Reviewed in [128]). As clutch size in XY females did not differ from XX siblings and some crosses from XY females have normal viability, it seems unlikely that widespread meiotic failure occurs in XY female stickleback. Furthermore, it is clear that Y chromosome transcripts or dosage differences from X chromosome genes do not prevent embryo development. Threespine stickleback are batch spawners that produce multiple clutches of eggs over a single breeding season [82]. If ovulated eggs are not passed, they can become overripe and inviable [129,130]. Overripe eggs can also harden and obstruct the fish, causing them to become eggbound. Many of the clutches from XY *amhy*-KO females had this hardened, overripe appearance that is uncommon in our lab-reared, wildtype fish. In addition, some XY females we attempted to cross had no cloaca dilation and were eggbound. We suspect that subfertility of XY females is not due to an inability to produce viable eggs, but due to an inability to pass these eggs once mature. Whereas wildtype fish can pass eggs on their own and begin another cycle of ovulation, XY females retain these eggs until they are manually removed. Repeated and frequent crossing would prevent overripe eggs from building up and allow the harvest of freshly ovulated, viable eggs. Hormones produced by the gonad and pituitary gland appear to play a role in egg maturation and overripening for stickleback [130,131], so while XY females are histologically similar to XX females, variation in hormone production could inhibit proper production and release of eggs.

The mammalian Y chromosome has several genes essential for spermatogenesis and sperm development [132–136]. Single cell sequencing has shown that many Y-specific genes are expressed throughout spermatogenesis in stickleback [137], but these genes are apparently not essential for this process. Several XX *Tg(amhy,EGFP)* males from this study had testes with many fully mature spermatozoa that were able to fertilize eggs and generate viable progeny at similar rates to wildtype sperm. This raises a question of the importance of the Y chromosome to fitness and reproduction. Testes from fertile XX males did not show an obvious reduction in sperm number indicating that widespread meiotic arrest does not occur in XX males. Stickleback are external fertilizers in which sperm competition frequently occurs [138]. Variation in sperm morphology [139] and motility [140] can contribute to reproductive success, so there may be differences in sperm

function that our *ex vivo* fertilization assays do not capture. Some XX *Tg(amhy,EGFP)* males did have hypotrophic testes with a reduction in identifiable germ cells, but since this phenotype was not universal, we suspect that variation in transgene copy number and insertion location led to overexpression of *amhy* in these fish. The germline *Tg(amhy,EGFP)* offspring we produced will allow us to determine if this phenotypic variation exists in a consistent genetic background.

## A four core genotypes stickleback model

Sex chromosomes impact a wide range of traits outside of fertility that are still critical to fitness. By comparing XY males and XX females to sex-reversed XY females and XX males, the Four Core Genotypes mouse model has allowed researchers to dissect the role of genotypic and gonadal sex to a number of behavioral and physiological sex differences in mammals (reviewed in [141,142]) including mating behavior [143] and parental care [144]. The creation of reproductively mature XX males and XY females in this study offers a new vertebrate model to study the separate contributions of gonadal sex and sex chromosomes to sexual dimorphism.

Many traits are sexually dimorphic in the threespine stickleback, including shape and size of the head and body [43–46], bony armor features [43,44], and brain size [47,48] as well as feeding behaviors [49,50] and boldness [51]. Male stickleback show complex courtship and mating behavior [52,53] and male-only parental care [54], and females show a variety of mate choice preferences [55–60]. Male nuptial coloration is a particularly interesting trait, as strong coloration is beneficial for courtship [60,61], but has a cost of increased risk of predation [62,63]. Therefore, sexually antagonistic selection would select for stronger, more conspicuous coloration in males and inconspicuous coloration in females. Androgens, secreted from the gonads, are shown to influence nuptial coloration in stickleback [64]. These observations are supported by the presence of nuptial coloration in XX males in our study, which lack all Y-linked genes, but have functioning testes. In contrast, male nuptial coloration in XY females that lack testes shows that there are also sex-linked genetic factors that contribute to nuptial coloration independent of gonadal sex. Future research with the Four Core Genotypes stickleback will provide many insights into the genetic basis of sexual dimorphism and how sexually antagonistic selection shapes the evolution of sex chromosomes.

## Methods

### Ethics statement

All animal procedures were approved by the University of Georgia Animal Care and Use Committee (protocols A2021 07-031-Y3 and A2024 08-009-A12).

### Animal husbandry

All experiments used lab-derived progeny from wild-caught threespine stickleback fish from Port Gardner Bay (Washington, USA) and Japan Pacific Ocean (Akkeshi, Japan). We collected fish from Washington State over multiple years using Scientific Collection Permit numbers 21-122, 22-150, 23-147, and 24-143. All experiments were performed using fish crossed and reared in laboratory conditions. Fish were maintained in 3.5 ppt Instant Ocean Sea Salt (Spectrum Brands, Blacksburg, VA, USA) in reverse osmosis water at 18 °C and pH 8.0 and a summer photoperiod (16L:8D). After hatching, fry were fed freshly hatched Grade A Brine Shrimp (Brine Shrimp Direct, Ogden, UT, USA) twice daily.

### CRISPR/Cas9 knockout of *amhy*

We microinjected stickleback embryos following standard procedures [41], with the addition of KCl to the injection mix at a final concentration of 0.2 M [145]. Cas9 protein was purchased from QB3 MacroLab (University of California, Berkeley). Synthetic sgRNAs were designed using CHOPCHOP [146–148] and purchased from Synthego: amhy_sgRNA_77r (5'-GATCAGCTGCTGTCCATGCA-3') and amhy_sgRNA_600r (5'- GGTCCTCTCTCGTCTTCGTT -3'). We combined Cas9

protein and sgRNA at a final concentration of 10 µM each and incubated for at least ten minutes at room temperature to allow the ribonucleoprotein (RNP) to form. The injection mix was made to a final concentration of 4 µM RNP.

## Synthesis of constructs and transgenesis

Plasmids pBT2 [149], pUbi:Switch [75], and pCS-zT2TP [8] were kindly shared by Mary Goll. Plasmid pT2AL200R150G [71] (referred to as pT2Xla.Ef1a:EFGP) was kindly shared by Tyler Square and Craig Miller. We purchased a synthetic *amhy*,EGFP construct from TWIST Bioscience (San Francisco, CA, USA). This construct consisted of a Kozak consensus sequence (5'-GCCGCCACCATGG-3'), N-terminus flag tag, the threespine stickleback *amhy* coding sequence, a P2A self-cleaving peptide [73,74], EGFP, and an SV40 poly(A) sequence, all flanked by 5' BamHI and 3' NotI restriction sites to facilitate cloning. We performed all restriction digests following the manufacturer's guidelines (New England Biolabs). We isolated all of the plasmids using a Monarch Plasmid Miniprep Kit (NEB #T1010) and gel purified all DNA fragments using the Monarch Gel Extraction kit (NEB #T1020). DNA fragments were ligated using T4 DNA Ligase following manufacturer recommended protocols (NEB #M0202). We transformed all plasmids into DH10B competent cells (Thermo Scientific EC0113). All plasmids were sequenced by Azenta Life Sciences (South Plainfield, NJ, USA) (S2 File).
   We assembled each construct as follows (S6 Fig):

1. pT2Dre.Ubi:amhy,EGFP: The *D. rerio* ubiquitin promoter was excised from pUbi:Switch and cloned into pBT2 using XhoI and BamHI. The synthetic *amhy*,*EGFP* construct was excised from pTwist_amhy-P2A-EGFP and cloned into the intermediate pT2Dre.Ubi backbone using BamHI and NotI.

2. pT2Dre.Ubi:EGFP: the *D. rerio* ubiquitin promoter was excised from pUbi:Switch and cloned into pT2Xla.Ef1a:EGFP using XhoI and BamHI.

3. pT2Xla.Ef1a:amhy,EGFP: the synthetic *amhy*,*EGFP* construct was excised from pT2Dre.Ubi:amhy,EGFP and cloned into pT2Xla.Ef1a:EGFP using BamHI and BglII.

4. pT2-5amhy2.5,Xla.Ef1a:EGFP: an 11.5 kb region of the Y chromosome containing *amhy* along with 5 kb of upstream and 2.5 kb downstream DNA was excised from bacterial artificial chromosome STB26-N21 from the CHORI-215 library [150] and cloned into pBT2 using BamHI and XhoI. The Xla.Ef1a:EGFP cassette was excised from pT2Xla.Ef1a:EGFP by digestion with XhoI and HpaI and ligated into the intermediate pT2-5amhy2.5 plasmid following digestion with XhoI and SwaI.

   We synthesized Tol2 mRNA using the mMessage mMachine SP6 Transcription kit (Invitrogen AM1340) and purified the product with the MEGAclear Transcription Clean-Up Kit (Invitrogen AM1908). We injected stickleback embryos as in [41] with the addition of KCl to a final concentration of 0.2 M [145]. All embryos were screened for EGFP expression five to seven days after injections before hatching had occurred. We removed all embryos without discernible somatic EGFP fluorescence.

## Genotyping and sanger sequencing

We extracted DNA from fin clips or whole embryos using a HotSHOT isolation protocol [151]. We performed all PCR reactions on an Analytik Jena Biometra Tone in a total volume of 20 µl using 0.8 µL of genomic DNA, 0.2 µM each primer, 0.2 mM each dNTPs, 0.5 units DreamTaq DNA polymerase (Thermo Scientific EP), and 2 µl 10x DreamTaq Green Buffer with 20 mM MgCl$_2$. We genotyped fish for sex using the sex-linked marker *idh* [39] with the following cycling conditions: 1 cycle of 95°C for 2 min; 30 cycles of 95°C for 30 s, 57.3°C for 30 s, and 72°C for 30 s; 1 cycle of 72°C for 5 minutes. All other reactions used: 1 cycle of 95°C for 2 min; 35 cycles of 95°C for 30 s, 60°C for 30 s, and 72°C for 30 s; 1 cycle of 72°C for 5 minutes. Primer sequences for all reactions are listed in S7 Table. We purified the PCR product using a Monarch PCR & DNA cleanup kit (NEB #T1030). All samples were Sanger sequenced through Azenta Life Sciences (South Plainfield, NJ, USA).

## Phenotyping and crossing

To phenotype gonads, we euthanized adult stickleback fish in 0.05% MS-222 buffered to neutral. We dissected out the gonads and fixed one or both gonads for 24–48 hours in 10% neutral buffered formalin. The remainder of the fish was preserved in 75% ethanol. The gonads were processed, embedded in paraffin, sectioned at 5 μm, and stained with hematoxylin and eosin by the University of Georgia College of Veterinary Medicine Histology Lab. We used a Zeiss Axio Scope A1 microscope and Zeiss Axiocam 305 Color camera to capture all images at the University of Georgia Biomedical Microscopy Core.

To make crosses, we euthanized mutant or wildtype male stickleback and gently macerated testes in 500 μl of Hanks' Balanced Salt Solution. We stripped eggs from gravid females and applied approximately 50 μl of sperm solution with a disposable transfer pipette.

We photographed adult stickleback with a Samsung Galaxy S22 under standardized lighting. We quantified coloration using FIJI (v. 1.54p) [152]. We measured total fish body area using the SIOX: Simple Interactive Object Extraction plugin. We used the Threshold Color plugin to measure blue ($Y = 0 - 254$, $U = 100 - 255$, $V = 0 - 131$) and dark ($Y = 0 - 31$, $U = 0 - 255$, $V = 0 - 255$) body area in masked images. Pixel counts are presented in S3 File. We calculated the proportion of body area for each color and performed a beta regression for differences in means in R (v. 4.4.1) and RStudio (v. 2024.04.2 + 764) with the packages betareg (v. 3.2-4) and emmeans (v. 1.11.2-8), allowing dispersion to vary by genotype (color ~ genotype | genotype).

## Riboprobe synthesis

We generated riboprobe templates for *gsdf, dmrt1, foxl2,* and *cyp10a1a* from cDNA from adult gonads. We extracted total RNA from mature testis and ovary tissue using TRIzol Reagent (Invitrogen 15596026) according to manufacturer's instructions. We synthesized cDNA using the iScript cDNA Synthesis Kit (Bio-Rad 1708891) and amplified the targeted region for each gene through PCR as described above (S7 Table). We cloned the PCR product into plasmid pJC53.2 (kindly shared by Kendall Clay and Rachel Roberts-Galbraith) after digestion with Eam1105I (Thermo Scientific FD0244) as in Collins *et al.* 2010 [153]. We verified each riboprobe template after cloning with sanger sequencing by Azenta Life Sciences (South Plainfield, NJ, USA). For *amh*, we ordered a synthetic gene fragment from TWIST Bioscience (San Francisco, CA, USA) containing a 307 bp region of the *amh* cds flanked by a 5' T3 promoter and 3' T7 promoter to facilitate riboprobe synthesis (S2 File).

To synthesize riboprobes, we PCR amplified linear template from the appropriate plasmid or used the synthetic gene fragment directly. We performed *in vitro* transcription from these templates using T7 (New England Biolabs M0251S), T3 (New England Biolabs M0378S), or SP6 (New England Biolabs M0207S) RNA polymerase according to manufacturer's instructions, substituting 0.2 mM of UTP for DIG-labeled UTP (Roche 11209256910). We purified riboprobes with a DNase treatment (New England Biolabs M0303S) and an ammonium acetate precipitation.

## *In situ* hybridization

We adapted a colorimetric *in situ* hybridization (ISH) protocol from Square, *et al.* 2021 [154]. We fixed the dissected gonads for 24–48 hours in 10% neutral buffered formalin. The gonads were processed, embedded in paraffin, and sectioned at 5 μm by the University of Georgia College of Veterinary Medicine Histology Lab. Unless otherwise stated, we performed all steps in Lock-Mailer microscope slide jars in a volume of 9-11mL and at room temperature. To deparaffinize, we warmed the slides in a 65°C incubator for five minutes, allowed them to cool to room temperature, then washed the slides in Hemo-De once for 5 minutes then again for 10 minutes. We performed prehybridization washes indicated with an asterisk in a Tissue-Tek II Slide Staining Station rather than Lock-Mailer jars: 100% EtOH for 5 minutes*, 80% EtOH for 10 minutes*, Milli-Q water for 10 minutes*, PBST for 5 minutes*, 15 ug/mL proteinase K in PBST for 5 minutes, PBST rinse*, 4% PFA for 20 minutes, PBST twice for 10 minutes each*, 67°C pre-heated hybridization buffer (50% formamide, 5X SSC,

0.1% Tween, 5mg/mL CHAPS, 0.1mg/mL yeast RNA, 0.1mg/mL heparin, pH 6.0 with citric acid) twice for 5 minutes each, and pre-heated hybridization buffer for 1–4 hours on a rotator at 67°C. We performed hybridization with 100–500 ng/mL of riboprobe in hybridization buffer incubated overnight rotating at 67°C. We stored hybridization solution at 20°C and reused up to 3 times.

Post-hybridization, we washed the slides 6 times with hybridization wash (50% formamide, 5X SSC, 0.1% Tween) rotating at 67°C for 20–90 minutes each, totaling 5–6 hours. We rinsed the slides in MABT (maleic acid buffer with Tween) and washed them in MABT twice for 20 minutes each, first at 67°C, then at room temperature. We added blocking solution (2% Blocking Reagent (Roche 11096176001) in MAB) to each slide, covered the slides with parafilm, and incubated for 1–3 hours in a humidor. We poured off the blocking solution and added 1:2000 anti-digoxigenin alkaline phosphate antibody (Roche 11093274910) in blocking solution, covered the slides with parafilm, and left them in a humidor overnight at 4°C. We performed five 20–50-minute post-antibody washes hybridization wash solution (50% formamide, 5X SSC, 0.1% Tween, citric acid to pH 6.0) on a rotator for a total of three to four hours, followed by an overnight MABT wash at 4°C.

Prior to coloration, we performed three 5-to-10-minute washes in NTMT (0.1 M NaCl, 0.1 M Tris pH 9.5, 0.05 M MgCl2, 0.1% Tween). We moved the slides into coloration solution (blocking solution plus 25 ug/mL NBT and 175 ug/mL BCIP) and incubated 2–30 hours covered to block light. We removed the slides from coloration solution once the color was stable in the positive control (*gsdf*), but before any background color was seen in the negative control (sense *gsdf*). Once coloration was complete, we rinsed the slides with PBST and washed for 10 minutes in PBST. We fixed the slides in 4% PFA in PBS at 4°C for one to five days.

To prepare slides for mounting, we performed three 5-minute washes in PBST and three 5-minute washes in deionized water. We mounted and coverslipped the slides using Vectashield antifade mounting medium with DAPI. We used a Zeiss Axio Scope A1 microscope and Zeiss Axiocam 305 Color camera to capture all images at the University of Georgia Biomedical Microscopy Core.

## Sequence alignments

We retrieved sequences for *amh* (Gene ID: 120823390) and *amhy* (Gene ID: 120812167) from threespine assembly GAculeatus_UGA_version5 (GCF_016920845.1) and *amh* (Gene ID: 119216504) from ninespine assembly fPunPun2.1 (GCF_949316345.1). We performed all alignments and calculated sequence identity between *amh* homologues with Muscle 5.1 [155] within Geneious Prime (v2024.0.5) using default parameters. We defined intron and exon positions based upon the threespine *amh* sequence in each alignment. We calculated dN/dS with the R package ape v5.8-1 [156] using the translation aligned coding sequences of *amh* (XM_040183306.1)*, amhy* (XM_040167960.1), and ninespine *amh* (XM_037469420.2). One additional base was included at the 5' end of exon 7 of the ninespine *amh* cds in order to correct the predicted annotation which contains multiple premature stop codons in this final exon. Alignment files are presented in S2 File. We generated Vistaplots [157,158] using LAGAN [159] alignments and a 100 bp window size with a 70% sequence identity threshold.

## RNAseq

We staged embryos according to established stickleback developmental stages [85], preserved the whole embryos in RNAlater Stabilization Solution (Invitrogen AM7021) following manufacturer guidelines, and stored at -80 °C until analysis. We extracted RNA using TRIzol Reagent (Invitrogen 15596026) and the Direct-zol RNA Microprep Kit (Zymo R2062). We created sequencing libraries using the Illumina TruSeq Stranded mRNA Library Prep kit (20020594). We pooled the libraries from all samples and 150 bp paired end reads were sequenced on two lanes using a NovaSeq X Plus by Azenta Life Sciences (South Plainfield, NJ, USA). All of the samples were sequenced to an average of 34.7 M read depth (min 11.6 M, max 71.3 M, SD = 10.4 M) (S1 File). We trimmed raw reads for quality and sequencing adapters with Trimmomatic

(v. 0.39) [160] using leading and trailing quality threshold of 3, a sliding window of 4 with quality threshold of 15, and minimum length cutoff of 36. We aligned paired and orphan reads to the threespine stickleback reference genome (GAculeatus_UGA_version5, NCBI RefSeq assembly GCF_016920845.1) using HISAT-3N [161] with default parameters except for reversing RNA strandedness (R for Read 1 and F for Read 2). All samples had an alignment rate between 66% and 87%, counting only uniquely mapped reads (S1 File). We generated Indexed BAM files of the alignments with Samtools (v. 1.17). We generated transcript counts against the NCBI annotation GTF file (release 100) for GCF_016920845.1 with HTseq (v. 2.0.2) [162] using HTSeq-count with default parameters except for reversing RNA strandedness (R for Read 1 and F for Read 2). We performed all remaining analyses in R (v. 4.4.1) and RStudio (v. 2024.04.2 + 764). We combined the counts from paired and orphaned reads for each sample and used DESeq2 (v. 1.44.0) [163] to generate normalized read counts and perform differential expression analysis. We identified DEGs using the standard workflow with stage and sex combined as a single factor. We used the contrast function to calculate log2 fold changes (logFC) and adjusted p values for differential expression between males and females at each timepoint. We identified DEGs of interest using GO terms from the NIH gene2go database (https://ftp.ncbi.nlm.nih.gov/gene/DATA/).

## Supporting information

**S1 Fig.  Sequence conservation of the *amhy* transgene region with autosomal *amh*.** Vistaplot showing conserved sequence identity of *amhy* to *amh* with sequence identity on the Y axis and base pair position of *amhy* on the x axis. Exon positions are indicated in dark blue, and the amh domain and TGF-β domain are indicated in light blue above the graph. Conserved sequences share 70% sequence identity in a 100 bp window.
(TIF)

**S2 Fig.  F0 *amhy* crispants show genetic mosaicism and variable sex reversal.** A) XY crispants developed as females with ovaries or males with testes. B) Sanger sequencing of exon 1 and exon 3 show various mutations relative to wildtype. Discordant chromatograms in crispants are indicative of mosaic mutations. Mutations are bolded in orange. CRISPR target site and PAM sequence are indicated by orange lines above wildtype reference sequence.
(TIF)

**S3 Fig.  Sanger sequencing of four F1 *amhy* mutant alleles.** Inserted base pairs are bolded. Locations of deletions are indicated with red arrowheads.
(TIF)

**S4 Fig.  Expression of gonadal genes in wildtype and sex-reversed ovaries.** *In situ* hybridization of gonadal genes in three biological replicates for wildtype XX ovaries and XY *amhy*-KO ovaries. Scale bars = 200 μm.
(TIF)

**S5 Fig.  CRISPR/Cas9 sgRNAs for *amhy* have multiple mismatches to their homologous site in *amh*.** PAM and sgRNA sequences are shown above the reference sequence. Mismatched nucleotides are bolded in orange text. Sanger sequencing of sex-reversed XY F1 *amhy*-KO fish showed no off-target mutations in *amh* in exon 1 (A) or exon 3 (B).
(TIF)

**S6 Fig.  Plasmid maps of transgenes constructed in this study.** Transgenes used the ectopic *X. laevis* Ef1a or *D. rerio* Ubi promoter or the endogenous *amhy* promoter. Control transgenes express only EGFP. Other transgenes contain the *amhy* coding sequence or the full *amhy* sequence including introns. To screen for transgene integration, *amhy* transgenes include an EGFP joined to *amhy* with a P2A self-cleaving peptide or separate EGFP cassette. All transgenes are flanked by Tol2 transposon arms to facilitate transgene integration.
(TIF)

**S7 Fig. Expression of gonadal genes in wildtype and sex-reversed testes.** *In situ* hybridization of gonadal genes in three biological replicates for wildtype XY testes and XX *Tg(amhy,EGFP)* testes. For one wildtype replicate, the tissue was lost during staining for *cyp19a1a*. Scale bars = 200 µm.
(TIF)

**S8 Fig. Analysis of nuptial coloration in wildtype and sex-reversed stickleback.** Representative high and low coloration individual for XX wildtype females, XX *Tg(amhy,EGFP)* males, XY *amhy*-KO females, and XY wildtype males. Blue and dark regions were quantified for each individual. Coloration proportion values are shown below each individual.
(TIF)

**S9 Fig. PCA plots of normalized gene expression counts.** Principal components 1 and 2 with the Proportion of Variance Explained on the X and Y axes, respectively. When analyzing all genes or autosomal genes only, samples group by developmental stage and not sex. X chromosome genes show separation by both stage and sex. Y chromosome genes group primarily by sex with male samples separate by developmental stage.
(TIF)

**S10 Fig. Volcano plots of female to male differential expression by stage.** Positive log2 Fold Change values indicate higher expression in females, and negative values indicate higher expression in males. Significant genes (p < 0.05) are shaded. The number of male biased, female biased, and total differentially expressed genes are shown above each plot.
(TIF)

**S1 Table. Viability and Hatching rates of crosses between wildtype XX females and wildtype XY males.**
(DOCX)

**S2 Table. Viability and Hatching rates of crosses from F1 XY *amhy*-KO females with wildtype XY males.**
(DOCX)

**S3 Table. Viability and Hatching rates of crosses from F2 XX or XY *amhy*-KO females a wildtype XY male.**
(DOCX)

**S4 Table. Wald z-test statistics of estimated marginal means from beta regression of the effect genotype on nuptial coloration.**
(DOCX)

**S5 Table. Sequence conservation of the threespine *amh* to threespine *amhy* and ninespine *amh*.**
(DOCX)

**S6 Table. Combined *amh* and *amhy* expression show significant differences between males and females at later stages.** Length adjusted *amhy* counts were calculated by multiplying *amhy* normalized read counts by transcript length of *amh* (3895 bp) divided by *amhy* (2496 bp). P values show two tailed Welch's t-test for a difference between *amhy* and one half *amh* expression in males at each stage and one tailed Welch's t-test for a greater total *amh* dosage in males than in females at each stage. Significant values are bolded (p < 0.05).
(DOCX)

**S7 Table. PCR primers used in this study.**
(DOCX)

**S1 File. RNA-seq summary and statistics.**
(ZIP)

**S2 File. Sequence files.**
(ZIP)

**S3 File. Coloration data.**
(TSV)

## Acknowledgments

This study was supported in part by resources and technical expertise from the Georgia Advanced Computing Resource Center, a partnership between the University of Georgia's Office of the Vice President for Research and Office of the Vice President for Information Technology.

The imaging data used in this publication was produced in collaboration with the Biomedical Microscopy Core at the University of Georgia.

We thank Koichi Kawakami (National Institute of Genetics, Japan) for use of the Tol2 elements in this study.

## Author contributions

**Conceptualization:** Matthew Treaster, Michael A White.

**Data curation:** Matthew Treaster.

**Formal analysis:** Matthew Treaster.

**Funding acquisition:** Matthew Treaster, Michael A White.

**Investigation:** Matthew Treaster, Jenny McCann, Kyra S. Solovei, Ryan J. Palmieri.

**Methodology:** Matthew Treaster, Jenny McCann, Michael A White.

**Project administration:** Michael A White.

**Resources:** Michael A White.

**Supervision:** Matthew Treaster, Michael A White.

**Visualization:** Matthew Treaster, Jenny McCann.

**Writing – original draft:** Matthew Treaster.

**Writing – review & editing:** Matthew Treaster, Jenny McCann, Michael A White.

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
