## [Decision Letter · Decision Letter 0]

17 Jul 2025

PGENETICS-D-25-00679

A Y-linked duplication of anti-Mullerian hormone is the sex determination gene in threespine stickleback

PLOS Genetics

Dear Dr. Treaster,

Thank you for submitting your manuscript to PLOS Genetics. It has been reviewed by three expert reviewers who clearly recognized the significance of the study. After careful consideration of the reviews, we agree with reviewer 2 that a more focused analysis of sex-specific amha and amhy expression, as well as the inclusion of key controls, is required for the paper to meet PLOS Genetics's publication criteria as it currently stands. Therefore, we invite you to submit a revised version of the manuscript that addresses the points raised during the review process. Please consider the points raised by all three reviewers in your revisions. 

Please submit your revised manuscript within 60 days Sep 15 2025 11:59PM. If you will need more time than this to complete your revisions, please reply to this message or contact the journal office at plosgenetics@plos.org. Please include the following items when submitting your revised manuscript:

We look forward to receiving your revised manuscript.

Kind regards,

Cecilia Moens

Academic Editor

PLOS Genetics

Pablo Wappner

Section Editor

PLOS Genetics

Aimée Dudley

Editor-in-Chief

PLOS Genetics

Anne Goriely

Editor-in-Chief

PLOS Genetics

**Journal Requirements:**

https://journals.plos.org/plosgenetics/s/submission-guidelines#loc-parts-of-a-submission

Potential Copyright Issues:

i) Please confirm (a) that you are the photographer of 5, or (b) provide written permission from the photographer to publish the photo(s) under our CC BY 4.0 license.

2) If any authors received a salary from any of your funders, please state which authors and which funders..

6)  Please ensure that the funders and grant numbers match between the Financial Disclosure field and the Funding Information tab in your submission form. Note that the funders must be provided in the same order in both places as well.  

7) Thank you for stating "RNAseq data is available in the NCBI SRA under BioProject PRJNA1248589. Raw and normalized read counts are available in the NCBI GEO under accession GSE296766." Please note that, though access restrictions are acceptable now, your entire minimal dataset will need to be made freely accessible if your manuscript is accepted for publication. This policy applies to all data except where public deposition would breach compliance with the protocol approved by your research ethics board. If you are unable to adhere to our open data policy, please kindly revise your statement to explain your reasoning and we will seek the editor's input on an exemption. 

8) Kindly revise your competing statement to align with the journal's style guidelines: 'The authors declare that there are no competing interests.'

**Reviewers' comments:**

Reviewer's Responses to Questions

**Comments to the Authors:**

Reviewer #1: Treaster and colleagues investigated the genetic basis for sex determination in threespine stickleback (Gac) fish. Amh has evolved independently as a sex determining gene in multiple vertebrate species. The amh gene has been duplicated onto the Y chromosome in the threespine stickleback. Amhy was mutated in XY Gac fish by CRISPR/Cas, resulting in completely sex-reversed male-to-females. Conversely, a genomic fragment of amhy introduced into XX Gac fish by transposons led to female-to-male sex reversed fish. Thus, amhy is both necessary and sufficient for sex determination in Gac fish. These mutants also provided the opportunity to explore fertility, YY embryos, and hormonal/sex chromosome-dependent phenotypes.

Pg 16, 399-402 regarding ancestral role of amh for germ cells. Transgenic mice overexpressing human AMH have primordial follicles at birth, but these are subsequently lost and the remaining somatic tissues form seminiferous tubule-like structures (Behringer 1990). Perhaps this can be included in the discussion.

In mice, Durlinger (1999), showed that Amh regulated the recruitment of primordial follicles.

Pg 8, 194-204. The authors attempted to generate amh overexpressing fish but were not successful at the level of GFP expression, leading them to speculate that extremely high and widespread expression of amh is cytotoxic. However, in mice overexpression of human AMH is compatible with life (Behringer 1990).

I am curious what is the knockout phenotype of amha Gac fish. I am not asking for the experiment, but it would be nice to have this discussed.

Some of the panels in some of the figures would benefit from a white balance of the background.

Behringer RR, Cate RL, Froelick GJ, Palmiter RD, Brinster RL. 1990. Abnormal sexual development in transgenic mice chronically expressing Müllerian inhibiting substance. Nature 345, 167-170.

Durlinger AL, Kramer P, Karels B, de Jong FH, Uilenbroek JT, Grootegoed JA, & Themmen AP (1999). Control of primordial follicle recruitment by anti-Mullerian hormone in the mouse ovary. Endocrinology 140, 5789–5796.

Reviewer #2: The threespine stickleback is a key model organism for studying evolutionary mechanisms. However, the genetic basis of sex determination in this species has remained unknown. In this study, the authors identify a Y-linked duplicate of the gene encoding anti-Müllerian hormone (amh), which they name amhy. Duplications of amh or its receptor, amhr, have independently evolved as master sex-determining genes in several vertebrate lineages, making amhy a strong candidate for the sex-determining gene in stickleback. To test this hypothesis, the authors used CRISPR/Cas9-mediated gene knockout and transgenic overexpression approaches. They found that amhy is necessary for male development, as all XY individuals lacking amhy developed as females. Conversely, many XX individuals with ectopic amhy expression showed apparently complete female-to-male sex reversal, indicating that amhy is also sufficient to initiate male development and production of functional testes. These findings strongly support the role of amhy as the primary sex-determining gene in threespine stickleback. The authors then examined the expression of amha and amhy during the developmental window when sex determination is thought to occur. Surprisingly, they found no significant upregulation of amhy relative to its autosomal paralog, nor a marked difference in total amh expression between males and females during this period. This suggests that sex determination may depend on subtle, cell-specific expression patterns or evolutionary changes in the protein-coding sequence. Finally, by analyzing sex-reversed lines, the authors argue that a classic sexually dimorphic trait—blue male nuptial coloration—is influenced both by Y-linked genetic factors and by hormones, independent of sex chromosome genotype. Overall, the data support the hypothesis that amhy is the primary sex determinant in the threespine stickleback. However, there are a few issues that need to be addressed before this analysis is complete.

Major comments:

There are two major issues that need to be addressed. First, there is a lack of controls in several of the data sets presented. These are detailed in the line-by-line critique below. Second, the timepoints chosen for the RNA-seq analysis were expected to cover the window when sex determination had initiated. However, the data shows little to no sex-specific expression of genes that are known sex-specific genes in a wide number of vertebrate species. This could be due to 1) the assumed timing of sex determination is inaccurate (see point below about rearing conditions), and/or 2) the method of analysis was not sensitive enough to detect differences. To this point, estimating gene expression levels from data generated from whole animal RNA will not be as accurate as using RNA isolated from only gonads. If it is not possible to isolate gonads from these early stages (it should be after some practice), then perhaps it would be more informative to analyze gene expression for the canonical sex-specific genes use RNA in situ on sections isolated gonads or on sections where gonads can be identified. In short, the conclusions drawn from the current gene expression analysis are the weakest part of this otherwise compelling manuscript.

Minor comment/suggestion:

It is not necessary to refer to autosomal amh as amha and could be interpreted to imply that there is an amhb. In other fish, “a” and “b” are reserved for ohnologs that resulted from the teleost-specific whole genome duplication. By contrast, amh and amhy are most likely paralogs that resulted from a small-scale duplication/translocation. Also, precedent in medaka to refer to autosomal dmrt1 as dmrt1 and Y-linked as dmy.

Lin-by-line comments/suggestions:

Line 64 This sentence is awkward.

Line 149: “…these distal portions OF amhy,…”; “…but PCR PRIMERS spanning…”

Line 172: Does “PG female” refer to Port Gardner Bay population fish? If so, then for clarity please define this is Line 132 when Port Gardner is first mentioned.

Lines 194-205: This is a negative result that does not add to the main results of the paper. I suggest cutting this section.

Line 211: “…hours after fertilization (haf).” Is it not more common to use “hours post-fertilization (hpf)?” OK if haf is standard nomenclature for sticklebacks.

Line 213: How do these fertility/survival rates compare to wild-type control crosses? Please add control cross numbers to Table S1.

Line 215: This sentence is awkward. Perhaps simply state that ¾ of the arrested fish are genotypically YY.

Line 224: “Subfertility also extended to second generation…” Based on the hypothesis, it would be expected that all sex reversed females would be sub fertile, regardless of generation. The point of this statement is therefore not clear.

Line 228: “Two XY F2 females were also eggbound…” There are many reasons fish get eggbound. Without a direct comparison to the frequency with which wild-type fish become eggbound (i.e. comparison to a control group), it is not clear what point the authors are trying to make.

Line 232 “…and four.” Typo. Likely needs to be deleted.

Line 234: Please add wild-type control numbers for comparison and also state numbers as “% viability (n= )” as this is the most relevant number to compare.

Line 236: “…somatic GFP expression…” Is amhy normally expressed in somatic tissue or is this expression due to influence of insertion site? amh is gonad specific in all other vertebrates.

Line 337: “histology of these males…” It is not clear what males are being analyzed. Male offspring that have somatic GFP expression?

Lines 230-243. This section seems out of place since rest of Fig. 3 is presented above (i.e. this section comes after Figure 4 presentation).

Figure 3: Please add wild-type XY age-matched testis for comparison with XX Tg(amhy;egfp) testes. Images are two small to assess stages of spermatogenesis (e.g. arrow to spermatogonia in Fertile panel is pointing to cells that appear to be outside the tubule/lobule, while spermatogonia would be expected to be within the tubule/lobule.

Line 251: This sentence is confusing because it starts off talking about mutants (“Neither amhy mutant…” but then appears to include transgenics [presumably the XX Tg(amhy,EGFP)]. Please make it clear what the genotypes of these fish are. Transgenics are usually not referred to as mutants unless the insertion site disrupts a gene.

Line 278: Estimating gene expression levels/timing of amhy vs. amha (or any gonad-expressed gene) using bulk RNA seq on RNA isolated from whole fish to is problematic. At the time their expression initiates, the gonads and expected to be a very small proportion of the fish, and only a small number of cells/gonad may be expressing these genes. A more accurate approach would be to assay expression by RNA in situ hybridization on isolated gonads or paraffin sections were gonadal cells can be identified (or RT-qPCR on isolated gonads). This is especially important given that at these time points there appear to be no detectable differences in the expression of amha in males vs. females (more on this below).

Line 282: It is stated that the sex determination likely begins before 3-4 days post hatch (or 11-12 dpf) since this is when differences in germ cell between males and females. However, referenced paper [81] shows detectable differences only starting at 15 dpf (reared at 19.5ºC), which would be ~7 dph. This later timing is consistent with reported increase in ddx4 and piwil1 starting at 8 dph, which is more consistent with sex determination occurring between 4-8 dph (not prior to 4 dph, as stated). For better comparisons between these studies, it will be necessary to know the details of the rearing conditions. Please add this to the M&M (water type, temperature, food, etc... Refer to Ref 81 for example). Also, please consider using dpf instead of dph since this does not rely on the reader knowing when hatching occurs.

Lines 305-346: It is a curious result that neither the classic male (dmrt1) or female (foxl2a) genes show sex biased expression at the time points assayed (≤8dph), a time point the authors presume is after initiation of sex determination. However, an alternative hypothesis is that under the rearing conditions used, sex determination has yet initiate. It would therefore be necessary to extent the timepoints to ≥8dpf to a point where dmrt1 and foxl2a expression becomes sexually dimorphic. It may be that at these later timepoints, amhy is more strongly expressed in XY animals, as was predicted. It would also be informative to do RT-qPCR on isolated adult gonads to determine if these gene are expressed sex-specifically at some point. If they are, then this further supports that the timepoints analyze are not late enough to capture when sex determination is actually happening.

Line 342: “…hsd17b1, which plays a role in estrogen activation.” Should be “…estrogen synthesis.” as hsd17b1 converts E1 to E2.

Reviewer #3: In this study, the authors found that the sex-determining gene in the threespine stickleback is the amhy gene, which arose by copy-and-paste duplication from an autosomal copy. This result in itself is not particularly novel. However, the part where the authors manipulated this gene to produce XX males and XY females and showed that the Y chromosome is important for spermatogenesis and development of nuptial coloration and that YY individuals are lethal is extremely important. This is because the function of the Y chromosome other than sex determination remains largely unknown in many taxa, and it is not known whether sexually antagonistic genes (e.g., genes that favor males and disfavor females) accumulate on the Y chromosome or to what extent harmful mutations accumulate on the Y chromosome. Therefore, I recommend acceptance of this paper after proper revision.

One major suggestion is that the authors should explain these topics (functions and deleterious mutations of Y chromosomes) in Introduction. That will make the novelty of this paper more clear.

Here are other suggestions and questions in the order of appearance;

L248 "develop throats": "develop red throats"?

L254 "nuptial coloration": Which color do you mean, red throat, blue eye, or dark body?

L269 "less sequence identity": Can you show any number, such as % identity?

L271 "consistent with weaker purifying selection acting on the Y chromosome due to a lack of recombination": I cannot follow this logic. What data is consistent with weaker purifying selection?

L272-273: According to Table 1, which dN/dS value is higher appears to vary among regions.

Fig. 5: These are images of just representative individuals. Can you get any quantitative data?

**Have all data underlying the figures and results presented in the manuscript been provided?**

Reviewer #1: Yes

Reviewer #2: **No: ** There is a lack of controls for several data sets

Reviewer #3: Yes

PLOS authors have the option to publish the peer review history of their article (what does this mean? ). If published, this will include your full peer review and any attached files.

**Do you want your identity to be public for this peer review?** For information about this choice, including consent withdrawal, please see our Privacy Policy .

Reviewer #1: No

Reviewer #2: No

Reviewer #3: No

**Figure resubmission:**
---

## [Decision Letter · Decision Letter 1]

20 Oct 2025

Dear Dr Treaster,

We are pleased to inform you that your manuscript entitled "A Y-linked duplication of anti-Mullerian hormone is the sex determination gene in threespine stickleback" has been editorially accepted for publication in PLOS Genetics. Congratulations!

Yours sincerely,

Cecilia Moens

Academic Editor

PLOS Genetics

Pablo Wappner

Section Editor

PLOS Genetics

Aimée Dudley

Editor-in-Chief

PLOS Genetics

Anne Goriely

Editor-in-Chief

PLOS Genetics

BlueSky: @plos.bsky.social

Comments from the reviewers (if applicable):

Reviewer's Responses to Questions

**Comments to the Authors:**

Reviewer #1: The authors have sufficiently addressed my comments and those of the other referees.

Reviewer #2: The authors have addressed my major concerns. This is a much improved manuscript.

**Have all data underlying the figures and results presented in the manuscript been provided?**

Reviewer #1: Yes

Reviewer #2: Yes

PLOS authors have the option to publish the peer review history of their article (what does this mean? ). If published, this will include your full peer review and any attached files.

**Do you want your identity to be public for this peer review?** For information about this choice, including consent withdrawal, please see our Privacy Policy .

Reviewer #1: No

Reviewer #2: No

**Data Deposition**

http://datadryad.org/submit?journalID=pgenetics&manu=PGENETICS-D-25-00679R1

**Press Queries**

---

## [Editor Report · Acceptance letter]

PGENETICS-D-25-00679R1

A Y-linked duplication of anti-Mullerian hormone is the sex determination gene in threespine stickleback

Dear Dr Treaster,

We are pleased to inform you that your manuscript entitled "A Y-linked duplication of anti-Mullerian hormone is the sex determination gene in threespine stickleback" has been formally accepted for publication in PLOS Genetics! Your manuscript is now with our production department and you will be notified of the publication date in due course.

With kind regards,

Anita Estes

PLOS Genetics

On behalf of:
